# Organic Carbon Characteristics in Ice-rich Permafrost in Alas and Yedoma Deposits, Central Yakutia, Siberia

Torben Windirsch[1,2], Guido Grosse[1,2], Mathias Ulrich[3], Lutz Schirrmeister[1], Alexander N. Fedorov[4,5], Pavel Ya. Konstantinov[4], Matthias Fuchs[1], Loeka L. Jongejans[1,2], Juliane Wolter[1], Thomas Opel[1] and Jens Strauss[1]

[1]Alfred Wegener Institute Helmholtz Centre for Polar and Marine Research, Telegrafenberg A45, 14473 Potsdam, Germany
[2]University of Potsdam, Institute of Geosciences, Karl-Liebknecht-Straße 24-25, 14476 Potsdam, Germany
[3]Leipzig University, Institute for Geography, Johannisallee 19a, 04103 Leipzig, Germany
[4]Melnikov Permafrost Institute, SB RAS, 36 Merzlotnaya str., Yakutsk, Republic of Sakha, Russia, 677010
[5]BEST International Centre, North-Eastern Federal University, 58 Belinsky str., Yakutsk, Republic of Sakha, Russia, 677027

*Correspondence to:* Torben Windirsch (torben.windirsch@awi.de)

**Abstract.** Permafrost ground is one of the largest repositories of terrestrial organic carbon and might become or is already a carbon source in response to ongoing global warming. In particular, syngenetically frozen ice-rich Yedoma deposits originating from the late Pleistocene store a large amount of organic carbon (OC). After thaw, this carbon becomes available to the recent carbon cycle. With this study of Yedoma and associated Alas deposits in Central Yakutia, we aimed to assess the local sediment deposition regime and its effect on permafrost carbon storage. For this purpose, we investigated the Yukechi Alas area (61.76495° N, 130.46664° E), a thermokarst landscape degrading into Yedoma in Central Yakutia. We retrieved two sediment cores (Yedoma upland, 22.35 m deep, and Alas basin, 19.80 m deep) in 2015 and analyzed biogeochemistry, sedimentology, radiocarbon dates and stable isotope geochemistry. The laboratory analyses of both cores revealed very low total OC (TOC) contents (< 0.1 wt%) for a 12 meter section in each core, while the remaining sections ranged from 0.1 to 2.4 wt% TOC. Those core parts holding very little to no detectable OC consisted of coarser sandy material estimated to an age between 39,000 and 18,000 years before present. For this period, we assume deposition of organic-poor material. Pore water stable isotope data from the Yedoma core indicated a continuously frozen state except for the surface sample, thereby ruling out Holocene reworking. In consequence, we see evidence that no strong organic matter (OM) decomposition took place in the sediments of the Yedoma core until today. The Alas core from an adjacent thermokarst basin was strongly disturbed by lake development and permafrost thaw. Similar to the Yedoma core, some sections of the Alas core were also OC poor (< 0.1 wt%) in 17 out of 28 samples. The Yedoma deposition was likely influenced by fluvial regimes in nearby streams and the Lena River shifting with climate. With its coarse sediments with low OC content (OC mean of 5.27 kg/m³), the Yedoma deposits in the Yukechi area differ from other Yedoma sites in North Yakutia that were generally characterized by silty sediments with higher OC contents (OC mean of 19 kg/m³ for the non-ice wedge sediment). Therefore, we conclude that sedimentary composition and deposition regimes of Yedoma may differ considerably within the Yedoma domain. The resulting heterogeneity should be taken into account for future upscaling approaches on the Yedoma carbon stock. The Alas core, strongly affected by extensive thawing processes during the Holocene, indicates a possible future pathway of ground subsidence and further OC decomposition for thawing Central Yakutian Yedoma deposits.

## 1 Introduction

Permafrost deposits represent one of the largest terrestrial carbon reservoirs. Perennial freezing largely prevents decomposition and preserves organic material. These permafrost soil conditions are found in the ground of approximately one quarter of the Northern Hemisphere's land surface (Zhang et al., 1999). The estimated amount of frozen and unfrozen carbon stored in the terrestrial permafrost region is 1330 to 1580 gigatons (Gt) (Hugelius et al., 2014; Schuur et al., 2015), which is approximately 45% more than what is currently present in the atmosphere (~ 864 Gt, based on 407 ppm $CO_2$ measured in 2018) (Ballantyne et al., 2012; Friedlingstein et al., 2019). Permafrost aggregation and conservation is highly dependent on long-term climatic conditions, both directly via air temperature and indirectly by the presence or absence of insulating vegetation and snow cover (Johansson et al., 2013). Currently, these permafrost conditions are under threat by rapidly increasing global, and in particular Arctic air temperatures which have resulted in widespread permafrost warming in recent years (Biskaborn et al., 2019). Gradual permafrost losses of up to 70% by 2100 in the uppermost 3 m are expected in a business-as-usual climate scenario (Chadburn et al., 2017; IPCC, 2019), and even deeper if accounting for deep thermokarst-induced rapid thaw (Nitzbon et al., 2020), while rapid permafrost thaw is not considered at all (Turetsky et al., 2020).

A special type of permafrost is the Yedoma ice complex deposit (in the following referred to as Yedoma), formed syngenetically by late Pleistocene deposition of fine-grained sediments with large volumes of ground ice. Yedoma is ice-rich (50–90 volume percent ice) and usually has organic carbon contents of 2 to 4 weight percent with an estimated deposit thickness up to 40 m (Schirrmeister et al., 2013; Strauss et al., 2013). In Central Yakutia, the cryostratigraphic characteristics of these syngenetic Late Pleistocene deposits have been previously studied by various researchers (Soloviev, 1959; Katasonov and Ivanov, 1973; Katasonov, 1975; Péwé et al., 1977; Péwé and Journaux, 1983). In the context of global climate change, such high ice content with intrasedimental ice and syngenetic ice wedges render Yedoma deposits highly vulnerable to thaw induced landscape changes (Schirrmeister et al., 2013) and ground volume loss causing surface subsidence. Thawing leads to ground subsidence that is often associated with thaw lake development (Grosse et al., 2013). Thaw lake development, surface subsidence, lake drainage, and refreezing of the sediments result in a thermokarst basin landform called Alas in Central Yakutia (Soloviev, 1973). During these thermokarst processes, the organic material stored within the permafrost becomes exposed to decomposition in the thaw bulbs (taliks) underneath the thermokarst lakes. It is subsequently released into the atmosphere as a result of microbial activity in unfrozen and aquatic conditions in form of gases such as carbon dioxide or methane, amplifying global climate change (Schuur et al., 2008). After a lake drainage event, the resulting thermokarst deposits in the Alas basins refreeze and the remaining Pleistocene soil carbon, as well as carbon from new plant biomass forming in thermokarst lakes and basins, becomes protected from decomposition again. The occurrence of these draining and refreezing processes can usually be determined by higher carbon content compared to the adjacent deposits (Strauss et al., 2013).

The resulting landscape patterns of Yedoma uplands and Alas basins form a heterogeneous landscape mosaic (Morgenstern et al., 2011). The heterogeneity and carbon characteristics within these deposit types, especially below 3 m, are still poorly studied, as only very few studies examining long Siberian permafrost cores have been conducted (Zimov et al., 2006; Strauss et al., 2013; Shmelev et al., 2017). Studies from this area mostly examine natural Yedoma exposures as for example in the Batagay mega thaw slump (Ashastina et al., 2017). In Central Yakutia, several permafrost studies have been conducted, especially on thermokarst processes, related surface dynamics and temperature changes (Fedorov and Konstantinov, 2003a; Ulrich et al., 2017a; Ulrich et al., 2017b; Ulrich et al., 2019). Other studies show a direct relation between dense vegetation cover and low permafrost carbon storage due to warmer permafrost conditions as a result of ground insulation (Siewert et al., 2015). Hugelius et al. (2014) estimate the carbon stock in the circumarctic permafrost region to be approximately 822 Gt carbon. However, despite the still high vulnerability of deeper deposits to thaw by thermokarst and thermo-erosion (Turetsky et al., 2019), only very few studies report organic carbon characteristics for permafrost deposits deeper than 3 m. This lack of data results in very high uncertainties for the impact of deep thaw in ice-rich permafrost regions and consequences for the carbon cycle (Kuhry et al., 2020).

By investigating deeper permafrost sediments in the continuous permafrost region of Central Yakutia, we aimed to understand the processes involved in organic carbon deposition and reworking in Yedoma and thermokarst deposits of this fast changing permafrost landscape (Nitze et al., 2018).

Our main research questions are: (1) What are the sedimentological processes that influenced the carbon stocks found in the Yedoma and Alas deposits of the Yukechi area?, and (2) How did the sedimentological processes affect the local carbon storage?

## 2 Study site

The Yukechi Alas landscape (61.76495° N; 130.46664° E) covers an area of approximately 1.4 km² and is located on the Abalakh Terrace (~ 200 m above sea level) in the Lena-Aldan interfluve of Central Yakutia (Fig. 1a) (Ulrich et al., 2019). It is characterized by Yedoma uplands and drained Alas basins indicating active thermokarst processes (Fedorov and Konstantinov, 2003a). Yedoma deposits cover 66.4 % of the area. The lakes cover about 13.0 % of the Yukechi Alas landscape, and approximately 20.6 % of the area consists of basins covered by grasslands, which contain Alas deposits (Fig. S1).

Today, Central Yakutia is characterized by an extreme continental subpolar climate regime with very low winter air temperatures down to minima of -63 °C in January (Nazarova et al., 2013). Holocene summer climate reconstructions indicate climate settings with slightly colder conditions ($T_{July}$ for 10,000 to 8,000 yr BP and 4,800 to 0 yr BP is $15.6 \pm 0.7$ °C,) compared to modern climate ($T_{July}$ is 16.6 to 17.5 °C) and a mid–Holocene warming phase between about 6,000 and 4,500 yr BP ($T_{July}$ ~ 1.5 °C higher than today) (Nazarova et al., 2013; Ulrich et al., 2017b). The contemporary mean annual air temperature in Central Yakutia (measured at Yakutsk Meteorological Station) is -9.7 °C. The modern active layer thickness in Central Yakutia is approximately 1.5 m but it can be larger in grasslands, such as within Alas basins (about 2 m and more), and smaller below the taiga forest (less than 1 m) (Fedorov, 2006). For the Yukechi Alas deposits, the active layer depth can be estimated at around 2 m and therefore reaches down into an observed talik, following Fedorov (2006). Taliks form because of a recent or already drained lake that prevented winter freezing, or an incomplete refreezing of the active layer.

The Yedoma deposits in this region can be more than 30 m in thickness as was already shown by older Russian works (Soloviev, 1959; 1973). Lakes are found in partially drained basins as well as on the surrounding Yedoma uplands (Fig. 1b). The land surface within the Alas basins is covered by grasslands while the boreal forest found on the Yedoma uplands mainly consists of *Larix cajanderi* with several *Pinus sylvestris* communities (Kuznetsova et al., 2010; Ulrich et al., 2017b). Central Yakutian Alas landscapes are characterized by extensive land use (mainly horse and cattle herding and hay farming) (Crate et al., 2017).

Lake dynamics have been monitored at the Yukechi Alas study site for several decades by the Melnikov Permafrost Institute in Yakutsk (Bosikov, 1998; Fedorov and Konstantinov, 2003b; Ulrich et al., 2017a) and have partially been linked to local land use (Crate et al., 2017).

Figure 1 – Study site overview; a: location of the Yukechi Alas study site in Central Yakutia on the edge of the Abalakh Terrace (Circumpolar digital elevation model, Santoro and Strozzi, 2012); b: locations of the Alas1 and the YED1 coring site within the Yukechi Alas landscape (Planet OrthoTile, acquisition date: 7 July 2018; Planet Team (2017)).

## 3 Methods

### 3.1 Field work

Field work took place in March 2015 during a joint Russian-German drilling expedition. Two long permafrost sediment cores were obtained, one from Yedoma deposits and one from the adjacent drained Yukechi Alas basin (Fig. 1b). The surface of the Alas sample site (61.76490° N, 130.46503° E; h = 209 m above sea level) is located approximately 9 m lower than the surface of the sampled Yedoma site (61.75967° N, 130.47438° E; h = 218 m above sea level) (Fig. 2). The distance between the two coring locations is 765 m. Both cores were drilled from dry land surface, kept frozen, and sent to Potsdam, Germany, for laboratory analysis. The Yedoma core (YED1) reached a depth of 22.35 m below surface (bs) and includes an ice wedge section from approximately 7.0 to 9.5 m bs. A talik section due to not completely refrozen active layer as identifies at 100 to 200 cm bs. The Alas core (Alas1) reached 19.80 m bs. A talik section was found in the Alas core reaching from approximately 160 down to 750 cm bs.

Figure 2 – Setting of the drilling locations for the Alas1 and YED1 cores showing distance and height difference between the locations (vertical scale exaggerated); the terms "Alas lake" and "Yedoma lake" are chosen after Ulrich et al., 2017a in accordance to the deposit type in which the thermokarst lakes are located; following Crate et al., (2017), the Yedoma lake can also be called "dyede" due to its development stage.

### 3.2 Laboratory analyses

The frozen cores were split lengthwise using a band saw and were subsequently subsampled. Each subsample consisted of approximately 5 cm core material. Subsamples were equally distributed along the cores. According to visual changes, we covered all visible stratigraphic layers and we sampled at least every 50 cm in order to capture specific sediment properties. The samples were weighed and thawed. Intrasedimental ice or, if the sediment was unfrozen during drilling, intrasedimental water was extracted using artificial plant roots (Rhizones) consisting of porous material with a pore size of 0.15 µm and applied vacuum. In order to avoid evaporation, the samples were thawed at 4 °C inside their sample bags and sealed tightly after inserting the Rhizones. These water samples were then analyzed for stable oxygen and hydrogen isotopes (see section 3.2.5). The ice wedge ice was subsampled using a saw for the analysis of stable oxygen and hydrogen isotopes.

### 3.2.1 Ice content, bulk density, and subsampling

The weighed sediment samples were freeze-dried and weighed again afterwards for determining the absolute ice content in weight percent (wt%). We decided for the absolute ice content as the gravimetric one, normalized with the dry sample weight, is not suitable for further calculations. Ice content within talik areas represents the water content, which froze after drilling. Bulk density was calculated from the absolute ice content, assuming an ice density of 0.9127 g/cm$^3$ at 0 °C and a mineral density of 2.65g/cm$^3$ (Strauss et al., 2012).

### 3.2.2 Elemental analyses

Subsamples used for elemental analyses were homogenized using a planetary mill (Fritsch Pulverisette 5). Subsamples were then weighed into tin capsules and steel crucibles for the elemental analyses. Total carbon (TC), total nitrogen (TN), and total organic (TOC) content were measured through combustion and analyses of resulting gases using a vario EL III and a varioMAX C Element Analyzer. Results give the carbon and nitrogen amounts in relation to the sample mass used for analysis in wt%. The carbon nitrogen ratio (C/N) was calculated from the TN and TOC content. Besides showing an input signal, we used this ratio as a rough indicator for the state of degradation or source of organic matter. Assuming a constant source, a higher ratio indicates better-preserved organic matter (Stevenson, 1994; Strauss et al., 2015).

### 3.2.3 Magnetic susceptibility and grain size analysis

Subsamples taken for grain size analysis were first measured for mass specific magnetic susceptibility using a Bartington Magnetic Susceptibility Meter Model MS2 and a frequency of 0.465 kHz. This allows us to differentiate between different mineral compositions (Butler, 1992; Dearing, 1999). Values are given in SI units (10$^{-8}$ m³/kg). For grain size analysis, the samples were treated with hydrogen peroxide and put on a shaker for 28 days to remove organic material. The pH was kept at a reaction-supporting level between 6 and 8. Subsequently, the samples were centrifuged and freeze–dried. Of each sample, 1 g of each sample was mixed with tetra-Sodium Pyrophosphate 10–hydrate ($Na_4P_2O_7*10H_2O$) (dispersing agent) and dispersed in an ammonia solution. The grain size distribution and proportions were determined using a Malvern Mastersizer 3000 equipped with a Malvern Hydro LV wet–sample dispersion unit. Statistics of the grain size measurements were calculated using Gradistat 8.0 (Blott and Pye, 2001). Results are used to identify different stratigraphic layers via material composition and to deduce sedimentary processes.

### 3.2.4 Radiocarbon dating

Radiocarbon dating was done for nine samples using the Mini Carbon Dating System (MICADAS) at AWI Bremerhaven. We used bulk sediment samples for dating due to a lack of macro-organic remains within the deposits. The results were calibrated with the software Calib 7.1 (Stuiver et al., 2018) using the IntCal13 calibration curve

(Reimer et al., 2013). Results are given in calibrated years before present (cal yr BP). The age–depth model was developed using the "Bacon" package in the R environment (Blaauw and Christen, 2011) (Fig. S2).

### 3.2.5 Stable isotopes

Besides showing a source signal (Meyers, 1997), stable carbon isotopes can be used as a proxy for the degree of decomposition of organic material, as during decomposition and mineralization 12C is lost, resulting in a higher share of 13C and hence a higher $\delta^{13}C$ ratio (Fig. S3) (Diochon and Kellman, 2008).

Twenty-three subsamples for $\delta^{13}C$ analysis were ground and carbonates were removed by treating the samples with hydrochloric acid for three hours at 97.7 °C. The samples were then vacuum-filtered, dried, and weighed into tin capsules for analysis. The stable carbon isotopes were measured using a Delta V Advantage Isotope Ratio MS supplement equipped with a Flash 2000 Organic Elemental Analyzer. The results are compared to the Vienna Pee Dee Belemnite (VPDB) standard and given in per mille (‰) (Coplen et al., 2006) with an analytical accuracy of $\leq 0.15$ ‰. Stable hydrogen and oxygen isotopes can be used as a temperature proxy. Lower $\delta^2H$ and $\delta^{18}O$ values indicate lower temperatures during precipitation. Samples taken from the ice wedges generally yield a winter temperature signal (Opel et al., 2018), whereas pore ice and pore water signals are a mix of different seasons with a higher uncertainty due to alteration and fractionation during deposition and multiple freeze-thaw cycles as well as evaporation (Meyer et al., 2000).

Our $\delta^2H$ and $\delta^{18}O$ samples were measured at AWI Potsdam Stable Isotope Laboratory using a Finnigan MAT Delta–S mass spectrometer with the equilibration technique after Horita et al. (1989). In total, 29 samples were measured, of which 16 originated from YED1 pore ice, 8 originated from YED1 wedge ice, and 5 from Alas1 pore ice or pore water. The results are given in per mille related to standard mean ocean water (‰ vs. SMOW). The analytical accuracy for $\delta^2H$ was $\leq 0.8$ ‰ and for $\delta^{18}O$ it was $\leq 0.1$ ‰ (Meyer et al., 2000). The d excess (d=$\delta^2H$-8*$\delta^{18}O$) was calculated as well from these values.

### 3.2.6 Statistics and bootstrapping approach for carbon budget estimations

For the mean grain size, the mean of each core unit, consisting of several samples' mean values, is given. We estimated the carbon budget of the Yukechi Alas area after Eq. (1), using a bootstrapping approach.

$$OC\ quantity\ (kt) = \frac{thickness*coverage*\frac{100-WIV}{100}*BD*\frac{TOC}{100}}{10^3} \tag{1}$$

with deposit thickness in m, coverage in m², WIV in vol %, BD in $10^3$ kg/m³ and TOC in wt%. For all TOC values below the detection limit (0.1 wt%), a value of 0.05 wt% was set. Missing bulk density values, resulting from low ice contents (< 20 wt%) (Strauss et al., 2012), were calculated after Eq. (2), which describes the relation between TOC and bulk density in the examined cores.

$$bulk\ density = 1.3664^{-0.115 * TOC} \tag{2}$$

The core length of the examined cores was assumed to represent the different ground types, resulting in a deposit thickness of 22 m for Yedoma deposits and 20 m for Alas deposits. A mean wedge-ice volume of 46.3 % for Central Yakutian Yedoma deposits and 7 % for Alas deposits of Central Yakutia was assumed following Ulrich et al. (2014).

We estimated the deposit coverage of Yedoma and Alas deposits using satellite imagery as shown in figure S1. The ice wedge in YED1 was excluded in the bootstrapping.

Bootstrapping calculations were done after Jongejans and Strauss (2020) for the upper 3 meters, the different core units as well as for the complete cores (Table 2) using the "boot" package in the R environment. Bootstrapping included 10,000 iterations of random sampling with replacement. We used combined BD and TOC values, as they are not

independent, and corrected for irregular sampling by value replication according to depth interval. We calculated the mean and standard deviation of all iterations.

## 4 Results

### 4.1 Characteristics of the Yedoma deposits

The Yedoma core YED1 visually appears rather heterogeneous (Fig. 3a) with material varying from fine gray material

(Fig. 3b[1]) to sandy grayish brown material (Fig. 3b[3]) (Windirsch et al., 2020b). Between 2235 and 1920 cm bs and between 691 and 0 cm bs, brown to black dots up to 2 cm in diameter may be organic-rich material. Cryostructures include structureless to micro-lenticular ice and larger ice veins and bands. The core penetrated an ice wedge between 1005 and 691 cm bs, so we could take ice samples only. The core contains an unfrozen layer close to the surface between approximately 200 and 100 cm bs, representing a thin initiating talik layer underneath the 100 cm thick frozen

active layer (Fig. 3a, red). All laboratory results are listed in detail in the PANGAEA repository (Windirsch et al., 2019).

**Figure 3 – a: overview of the Yedoma core; depth given in cm bs; state after core retrieval is given by colors: blue = frozen, red = unfrozen; location of the ice wedge is labelled; brown illustrates silty sediments, yellow represents sandy sediments; b: detailed pictures of the YED1 core; (1) 332–317 cm bs, picture of unit Y1 showing black organic-rich inclusions within**

25 **the grey silty matrix; (2) 960–944 cm bs, picture of the wedge ice in Y2; (3) 1549–1532 cm bs, picture of Y3, showing the coarse sandy material with no visible cryostructures or organic material; (4) 2133–2117 cm bs, picture of Y4, showing the grey silty matrix with some dark organic dots.**

We divided the Yedoma core into four main Yedoma units (Y) (Fig. 4). Y4 is the lowest (2235 to 1920 cm bs) and oldest (radiocarbon age of 49,323 cal yr BP) stratigraphic unit. The absolute ice content slightly increased towards the

30 surface (35.8 to 36.6 wt%, peak value of 53.6 wt% in between). MS also increased from 60.7 to 155.4 SI. The grain size was rather consistent with a mean value of $24.3 \pm 3$ µm and the soil texture varied between sand and silt (Fig. S4, S5). We found TOC contents of up to 1.7 wt% (mean of 1.3 wt%). The C/N ratios within this unit varied between 9.2 and 10.6, and $\delta^{13}C$ values ranged between -25.27 and -24.66 ‰ vs. VPDB (Fig. S3). TN values only reached the

detection limit of 0.1 wt% in 9 out of 36 samples in the whole YED1 core. As just these 9 samples exceeded the detection limit (highest value 0.16 wt% at 2036 cm bs), only they have been used for C/N ratio calculations.

The radiocarbon sample age of Y3 (between 1927 and 1010 cm bs) yielded an infinite age (> 49,000 yr BP) with $^{14}$C below detection limit. There is a transition zone between Y4 and Y3 represented by a diagonal sediment boundary in the core between 1927 and 1920 cm bs (see Fig. 3a). Y3 showed distinctly lower absolute ice contents (< 32.1 wt%). MS varied between 120.5 and 285.0 SI. Higher sand contents (> 56.9 vol%) led to an increase in grain size (72.1 to 191.6 µm) with a mean grain size of 120.5 ± 35.5 µm. Grain size decreased down to 33.3 µm in the uppermost sample of Y3 and no detectable TOC was found in this unit.

Y2 (1010 to 714 cm bs) consisted of massive wedge ice, which contained very little sediment inclusions (Fig. 3b [2]). Thus, only water isotopes ($\delta^2$H and $\delta^{18}$O) could be measured and analysed. The results are described in section 4.3.

Y1 (714 to 0 cm bs) is the uppermost and youngest unit with carbon ages ranging between 40,608 (589.5 cm bs) and 21,890 cal yr BP (157.5 cm bs). Ice content decreased from the ice wedge towards the surface ranging from 14.6 wt% (110 cm bs) to 57.4 wt% (688 cm bs). MS decreased towards the surface from 108.1 to 15.4 SI in the uppermost sample, with a maximum of 118.6 at 298 cm bs. This unit consisted of fine sediment with a mean grain size of 19.9 ± 4.2 µm. It contains up to 1.4 wt% TOC (298 cm bs). C/N values were in the range of 9.1 to 12.9. The lowest $\delta^{13}$C value was found at 21 cm bs with -28.07 ‰ vs. VPDB; the lower part of this section showed a mean value of -24.42 ± 0.6 ‰ vs. VPDB.

**Figure 4 - Characteristics of the Yedoma core YED 1: radiocarbon ages, absolute ice content, bulk density, magnetic susceptibility (MS), grain size composition, mean grain size, total organic carbon (TOC) content, carbon-nitrogen (C/N) ratio and stable carbon isotope ($\delta^{13}$C) ratio; hollow circle indicates an infinite radiocarbon (dead) age; grey/white areas mark the different stratigraphic units (Y1 to Y4).**

The grain size distributions (Fig. S5) illustrate the differences between the core units. Silt is the dominant grain size class in Y4 and Y1, whereas unit Y3 is dominated by sand.

The calibrated radiocarbon ages of the Yedoma deposits are listed in Table 1 and assigned to the different core units. Our age–depth model (Fig. S2a) indicates a steep age-depth relationship from approximately 1200 to 2235 cm bs and a rather well defined, gradual age-depth relationship from 1200 cm bs towards the surface (Fig. S2a).

The bootstrapping approach resulted in a mean soil organic carbon (SOC) estimation of 4.48 ± 1.43 kg/m³ for the top 3 m of the YED1 core and a mean of 5.27 ± 1.42 kg/m³ for the entire core (table 2). We calculated a carbon inventory of 56.8 ± 15.2 kt for the Yukechi Yedoma deposits by upscaling the carbon storage to the complete Yedoma coverage in the Yukechi Alas landscape (66.4 %, ~ 917,000 m²) (Fig. S1).

## 4.2 Characteristics of the Alas deposits

The Alas1 core contains a large proportion of unfrozen sediment (i.e. talik; ~ 750 to 160 cm bs) (Fig. 5a, red), which led to the loss of some core sections during drilling. The absolute ice content given for samples retrieved from this zone represents absolute water content; samples were frozen directly after core recovery and field description. The

core's visual appearance was more homogeneous compared to YED1 regarding color (greyish brown) and material (clayish silt (Fig. 5b[2]) to sandy silt (Fig. 5b[4])) (Windirsch et al., 2020a). Cryostructures of the frozen core below 750 cm bs included horizontal ice lenses up to 5 cm thickness and structureless non-visible ice. Blackish dots and lenses (up to 1 cm in diameter) hint that organic material is included in the sediments. The frozen sediment of the uppermost 160 cm bs represents the seasonally freezing layer.

**Figure 5 – a: overview of the Alas1 core; depth given in cm bs; state after core retrieval is given by colors: blue = frozen, red = unfrozen; light brown marks silty material, yellow marks sandy material, dark brown marks silty material containing more organic material; b: detailed pictures of the Alas1 core; (1) 88–64 cm bs, picture of A1 showing the silty grey matrix including dark organic structures; (2) 840–828 cm bs, picture of the sandy A2 unit; (3) 1169–1148 cm bs, picture of A3 showing a silty grey matrix with some darker organic dots; (4) 1781–1767 cm bs picture of the fine-grained silt-dominated A4 unit including black organic-rich inclusions.**

We divided the Alas1 core into four stratigraphic units (A1 to A4), according to soil texture and, if applicable, carbon content (Fig. 6). The oldest unit is A4 (1980 to 1210 cm bs) with radiocarbon ages of 42,865 cal yr BP (1967.5 cm bs) and 45,870 cal yr BP (1530.5 cm bs). An age inversion was detected here. Absolute ice content did not show a specific trend and ranged from 15.3 wt% at 1400.5 cm bs to 25.4 wt% at 1220 cm bs. MS ranged between 62.1 (1967.5 cm bs) and 133.9 SI (1759 cm bs) with much higher values in a sand intrusion found between 1530.5 and 1312 cm bs (266.7 at 1464 cm bs, 268.7 at 1400.5 cm bs). The mean grain size was constant ($35.9 \pm 36$ µm) except for the sandy intrusion (152.9 µm at 1464 cm bs, 72.6 µm at 1400.5 cm bs), leading to a high standard deviation (Fig. S4, S6). While TOC values were below detection limit within this sandy material, the other parts of A4 held TOC amounts of up to 1.8 wt% (1759 cm bs). The C/N ratio ranged between 5.8 (1274 cm bs) and 8.9 (1759 cm bs) with a mean value of 7.4 (Fig S2). The $\delta^{13}$C values showed a range of -25.67 to -24.06 ‰ vs. VPDB (Fig. S3). Only the TN values that exceeded the detection limit, which was the case in 8 out of 28 samples in the entire Alas1 core, have been used for C/N ratio calculations.

A3 ranged from 1210 to 925 cm bs. The absolute ice content was stable around $22.7 \pm 2.9$ wt%. MS increased towards the surface from 72.1 SI (1205.5 cm bs) to 122.6 SI (955 cm bs). A3 was characterized by less coarse material compared to A4 (Fig. S6), with a mean grain size of $19.7 \pm 3.7$ µm. All TOC values were below detection limit, so no C/N could be calculated and no $\delta^{13}$C could be measured.

The characteristics of A2 (925 to 349 cm bs) were similar to those of the sand intrusion found in A4. A radiocarbon age of 27,729 cal yr BP was measured at 812.5 cm bs. The absolute ice content had a mean of 15.2 wt% and decreased from 16.7 wt% at 919.5 cm bs to 12.9 wt% at 395 cm bs. MS decreased upwards from 302.3 to 129.2 SI. The mean grain size at the bottom of this unit was 102.3 µm (919.5 cm bs), increased to 221.9 µm at 812.15 cm bs towards the surface, and reached the lowest value of 41.2 µm at the upper boundary of A2 (Fig. S6) with an overall mean of $108.0 \pm 59.5$ µm. All TOC values were below the detection limit.

The uppermost stratigraphic unit A1 starts at 349 cm bs. It is the youngest unit of Alas1 with a radiocarbon sample at 199 cm bs dated to 15,287 cal yr BP. The absolute ice content slightly increased from 19.1 wt% (344.5 cm bs) to 23.1

wt% (9 cm bs) throughout this unit. MS decreased towards the surface, starting at 126.7 (344.5 cm bs) and reaching 50.8 at 9 cm bs. The mean grain size decreased again, compared to A2, representing silty material with values of $18.5^{+1.4}_{-1.6}$ μm. The mean grain size for this unit was 20.0 ± 4.6 μm. TOC was only detectable in the uppermost sample with a value of 2.4 wt% (9 cm bs). The C/N ratio for this sample was 12.0 and the $\delta^{13}C$ was -27.24 ‰ vs. VPDB.

**Figure 6 - Characteristics of the Alas1 core: radiocarbon ages, absolute ice content, bulk density, magnetic susceptibility (MS), grain size composition, mean grain size, total organic carbon (TOC) content, carbon-nitrogen (C/N) ratio and stable carbon isotope ($\delta^{13}C$) ratio; grey/white areas mark the different stratigraphic units (A1 to A4).**

The radiocarbon ages are listed in Table 1. The age-depth model (Fig. S2b) shows a rather continuous slope for all calibrated ages of Alas1.

Bootstrapping resulted in a mean SOC value of 6.93 ± 2.90 kg/m³ for the top 3 m of the Alas1 core (table 2). The calculation for the whole core resulted in a mean value of 6.07 ± 1.80 kg/m³ carbon. For the whole Alas area within the Yukechi Alas landscape (20.6 %, ~ 284,000 m²) (Fig. S1 [green]), we calculated a total organic carbon stock of 32.0 ± 9.6 kt using an estimated deposit thickness of 19.8 m.

## 4.3 Water isotope analysis of the YED1 and Alas1 core

Stable hydrogen and oxygen isotope results are shown in figure 7. We found clear downward trends of $\delta^{18}O$ and $\delta^2H$ values becoming more negative between 1000 and 400 cm bs in the YED1 core (Fig. 7b). Below 1000 cm bs, both $\delta^2H$ and $\delta^{18}O$ become less negative with increasing depth. $\delta^{18}O$ ranged between -25.16 ‰ at the lowermost sample and -30.70 ‰ at 1071.5 cm bs with a much less negative value of -15.53 ‰ closest to the surface. While the uppermost Yedoma sample had a $\delta^2H$ value of -120.8 ‰, all other Yedoma samples showed much more negative values between -181.3 ‰ (2209.5 cm bs) and -221.6 ‰ (1071.5 cm bs). Values almost aligned with the global meteoric water line (GMWL), but partly also the local evaporation line (LEL) of Central Yakutia (Wetterich et al., 2008), except for the ice wedge samples of YED1 (Fig. 7a). The isotope data obtained from the ice wedge samples had more negative values for both $\delta^2H$ (-220.6 ‰ to -228.6 ‰) and $\delta^{18}O$ (-29.58 ‰ to -30.55 ‰) in comparison to the remaining YED1 core. The d excess values are lowest in the YED1 ice wedge (lowest value of 9.3). Other values range between 3.5 in the uppermost sample as an outlier and generally range between 14.7 and 29.5 with no clear trend visible.

Most of the Alas samples were too dry to extract pore water for water isotope analysis, resulting in a low number of water samples for this core (Fig. 7c). These Alas1 samples showed little variance in $\delta^2H$ and $\delta^{18}O$ data, ranging from -13.33 ‰ (103 cm bs) to -15.48 ‰ (1154 cm bs) for $\delta^{18}O$ and -130.4 ‰ (61 cm bs) and -137.6 ‰ (1464 cm bs) for $\delta^2H$. d excess values are lower towards the surface (-22.8 at 61 cm bs, -24.3 at 103 cm bs) and range from -14.1 to -12.2 in the lower core part.

**Figure 7 – The characteristics of water stable isotopes in the studied sediment cores; a: stable hydrogen ($\delta^2H$) and oxygen ($\delta^{18}O$) isotope ratios of YED1 pore ice (black triangles), YED1 ice wedge ice (hollow triangles), and Alas1 pore ice and pore water (black dots) [‰ vs. SMOW]; global meteoric water line GMWL: $\delta^2H=8*\delta^{18}O+10$; local evaporation line LEL of Central Yakutia (based on data compiled until 2005 after Wetterich et al., 2008); b: oxygen isotopes, hydrogen isotopes and**

d excess values of YED1 plotted over depth; c: oxygen isotopes, hydrogen isotopes and d excess values of Alas1 plotted over depth.

## 5 Discussion

### 5.1 Carbon accumulation and loss at the Yukechi study site

We found surprisingly low TOC values in certain core sections of the Yedoma and Alas deposits. These low values appear in relation to coarser sediments (fine sand), while the rather fine sediment layers (silt and sandy silt) store more TOC. The similarities in sediment structure and composition of the two cores, in particular between units Y1, Y4 and A4 in terms of grain size composition and OC content, and the increased accumulation rates towards the core bottoms (Fig. S2) indicate that the sedimentary sources regime were the same for both cores until approximately 35,000 cal yr

BP (Fig. 4 and 6).

On the one hand, low TOC content could be caused by high organic matter decomposition during accumulation or during a thawed state, especially in thermokarst deposits. On the other hand, it could reflect low carbon inputs. A suitable explanation for a low-input scenario is a change in the sedimentary regime due to fluvial transportation processes as is explained in more detail in chapter 5.2. The low stable carbon isotope data of our cores (between -24.06

and -27.24 ‰) are comparable to other studied sites from the Yedoma domain (Schirrmeister et al., 2013; Strauss et al., 2013; Jongejans et al., 2018). Our C/N data suggest a more or less homogeneous source signal of the organic material. Both cores show the lowest $\delta^{13}$C values closest to the surface as the organic material is the most recent and therefore least decomposed. In deeper sections of the cores, $\delta^{13}$C is higher (less negative) with no general trend over depth in the Alas and Yedoma deposits. We interpret this as an indication of higher decomposed material already being

present before freezing. We see that decomposition ceased when the deposits became frozen, resulting in $\delta^{13}$C values not showing a clear trend at depth. The C/N ratios in both cores support this hypothesis and are in line with the results found by Strauss et al. (2015) and Weiss et al. (2016) for other Yedoma and Alas sites in Siberia. In comparison to the mean C/N ratio of 10 in YED1, the mean C/N ratio of 8 for Alas1 indicates that the Alas deposits were slightly more affected by decomposition due to their temporary thawed state during lake phase. As the carbon was freeze-

locked in the YED1 core the entire time since being frozen, it was not decomposed since deposition. The Yukechi C/N values are on the low end of C/N ratios known from other Yedoma deposits, e.g. from Bykovsky Peninsula (Schirrmeister et al., 2013) and Duvanny Yar (Strauss et al., 2012). The hypothesis of an input of organic-poor and already pre-decomposed material is rather supported by the fact that both cores, Alas1 and YED1, show low C/N ratios. The carbon characteristics indicate that the low carbon content results from low carbon input rather than high

decomposition rates in both cores, as no evidence for conditions favoring high decomposition rates was found. Therefore, low carbon content is likely not the result of high decomposition during the thawed lake phase rates during aquatic conditions of a lake-covered state, but a legacy of the source material. For a decomposition during lake phase

scenario, organic carbon parameters would differ largely in carbon content and isotope signature from those of the still-frozen Yedoma (Walter Anthony et al., 2014).

We found age inversions in both cores with similar age and depth (YED1 49,232 cal yr BP, 1998.5 cm bs; Alas1 42,865 cal yr BP, 1967.5 cm bs) (Fig. 4 and 6, Fig. S2) which is typical for many Yedoma sites (Schirrmeister et al., 2002). While cryoturbation might seem an obvious explanation, we suggest that this process did not play a major role here due to the long-term frozen state of YED1. Rather, we assume that the age inversions indicate a temporary shift in sediment input at approximately 35,000 cal yr BP. This could have caused some in-deposit reworking in the watershed and the incorporation of older material into younger sediments. In addition, the dating of bulk sediments very close to the maximum datable age of ~ 50,000 yr BP may cause a high uncertainty in the absolute ages of sediment layers (Reimer et al., 2013). Therefore, the rather small age inversions (> 49,000 cal yr BP to 49,232 yr BP in YED1, and 45,870 cal yr BP to 42,865 cal yr BP in Alas1) could be a result of material mixture in dated bulk samples. The radiocarbon ages above this age inversion align well with a simulated sedimentation rate, as shown in figure S2.

## 5.2 Yedoma and Alas development

The differences in ice content between both cores and the homogeneous ice content throughout the whole Alas1 core indicate that thaw processes influenced the Alas deposit. As described above, this is supported by the water isotope signals, which are quite homogeneous throughout Alas1. This is the quantitative evidence that these deposits have been previously thawed under thermokarst influence. The homogeneity in water isotopes is an outcome of percolating surface water during a thawed state. Subsequent talik refreezing in sandy sediments lead to the formation of structureless pore ice, forming a taberal deposit (Wetterich et al., 2009). Refreezing, in our case, started from the surrounding frozen ground rather than from the surface, as a talik is still present in the upper core part. This allowed for the formation of structureless, invisible to microlenticular ice structures in the sandy material providing relatively large pore spaces (French and Shur, 2010). Due to the formation of those small ice structures, no sediment mobilization by the formation of, for example, large ice bands occurred in this core, resulting in an unmixed and clearly layered sediment. This also excludes cryoturbational processes as an explanation for the age inversions that we found.

The perennially frozen conditions since incorporation into permafrost of the Yedoma deposits at YED1 are supported by the water isotope signals (Fig. 7) with much lower $\delta^{18}O$ values for the Yedoma pore ice in comparison to the uppermost sample (4 cm bs in YED1). The latter shows a water isotope signal reflecting very recent climate and freezing, thawing and evaporation processes in the active layer. If the Yedoma core had been thawed at some point, intruding water would have led to a more homogeneous oxygen isotope signal throughout the core as it is obvious in the Alas core. Also, the intact ice wedge gives evidence for a perennially frozen state throughout the depositional history at YED1. The stable isotope ratio values of wedge ice (mean $\delta^{18}O$ of -30 ‰, mean $\delta^2H$ of -224 ‰) reflects winter precipitation and fits well into the regional pattern for Marine Isotope Stage (MIS) 3 ice wedges in Central and Interior Yakutia (Popp et al., 2006; Opel et al., 2019) while the d excess shows a much elevated value (16 ‰) compared

to the regional pattern (Popp et al., 2006; Opel et al., 2019). The d excess values from the middle part of the ice wedge correspond well to the regional values from Mamontova Gora, Tanda and Batagay (Opel et al., 2019), while the others fit to those of the host sediments and are potentially overprinted by exchange processes between wedge ice and pore ice (Meyer et al., 2010). Due to the low number of datapoints, no meaningful co-isotopic regression was calculated.

The stable isotope composition of pore ice shows a co-isotopic regression of $\delta^2H = 6.61\ \delta^{18}O - 18.0$ (R² = 0.97, n = 23), which is typical for Yedoma intrasedimental ice (Wetterich et al., 2011; 2014; 2016). The isotope values plot well above the regional Local Meteoric Water Line of the cold season (Papina et al., 2017), suggesting a substantial proportion of (early) winter precipitation – usually characterized by high d excess values – for the pore ice, which is also evident for some units of the Batagay megaslump (Opel et al., 2019). The decreasing trend of pore ice isotopic $\delta$

values from the bottom to the top indicates a general cooling in Central Yakutia during the covered time span of our study. However, as it is accompanied by an opposite increasing trend in d excess, these values may be overprinted by secondary freeze-thaw processes in the active layer and rather reflect the intensity of these fractionation processes (Wetterich et al., 2014).

The age-depth models of both cores show steep curves and higher sedimentation rates at the bottom of both cores,
which slow down towards the surface (Fig. 4 and 6, Fig. S2). This indicates that during the early phase of the sediment accumulation (~ 45,000 to 35,000 cal yr BP), the depositional environment at Alas1 was the same as at YED1. The steepness of the age–depth model suggests an upward decrease in the accumulation rate or can be interpreted as an increase in surface erosion towards the top of the YED1 core (Figure S3a). Especially the sandy core part of Y3 accumulated rapidly, as indicated by the radiocarbon sample below dated to 49,232 cal yr BP (71.5 cm below the
bottom of Y3) and the next radiocarbon sample above dated to 40,608 cal yr BP (420 cm above the top of Y3). These 917 cm of Y3 therefore accumulated in less than 8,600 years, while in Y1, the accumulation of 714 cm took more than 18,700 years (40,608 cal yr BP at 589.5 cm bs, 21,890 cal yr BP at 157.5 cm bs) (Table 1). The continuous steepness of the age-depth model of Alas1 (Fig. S2b) suggests a rather constant accumulation rate throughout the deposition of these sediments.

Due to the alternation of coarse and carbon-poor material (i.e. Y3 and A2, see Fig. 4 and 6) with fine carbon-rich material (i.e. Y1 and Y4 in Fig. 4, A4 in Fig. 6), we suggest shifts in the sedimentary regime at the Yukechi study site (Soloviev, 1973; Ulrich et al., 2017a; Ulrich et al., 2017b). This hypothesis is supported by the MS results, which give higher values for sandy core parts, hinting at a different material source, compared to the silty and carbon-bearing core units. Due to the great thickness of those sandy layers (core units 1 to 4, Fig. 4 and 6), the most suitable explanation is
material transport by tributaries on top of the (former) Yedoma uplands of the Abalakh Terrace. We interpret that the sandy material in the studied cores was deposited during the river-connected flooding phases at our study site. Moreover, fluvial transport gives a suitable explanation for low carbon content as organic matter decomposition is often much higher under aquatic conditions (Cole et al., 2001). Furthermore, high flow velocity allow for larger

particles to be deposited, but keep lighter particles, like organic material, in suspension (Anderson et al., 1991; Wilcock and Crowe, 2003; Reineck and Singh, 2012).

Another explanation for the occurrence of these carbon-poor sandy layers are shifts in wind direction and wind speed and therefore the sediment carrying capacity of the wind (Pye, 1995). A shift in eastern Siberian climate during the beginning of the Kargin interstadial (MIS 3, ~ 50,000 yr BP) resulted in higher winter temperatures (Diekmann et al., 2017) and therefore higher pressure gradients within the atmosphere, leading to greater wind speeds. This in turn resulted in higher sediment carrying capacity of the wind which provides a suitable explanation for the sediment differences. Also, sand dunes of the Lena River valley (Huh et al., 1998) could have provided sufficient sandy material throughout the formation of the sand layers found in the Yukechi deposits (Y3 and A2 in Fig. 4 and 6). The radiocarbon ages of these coarser core segments (Y3 and A2 in Fig. 4 and 6) dated between 39,000 and 18,000 cal yr BP match the timing of these climatic changes. Increased wind speeds at the beginning of a warmer interstadial phase during the MIS 3 (Karginian climate optimum, 50,000 to 30,000 yr BP) and a subsequent decrease in wind speed during the colder stadial MIS 2 are a suitable explanation (Diekmann et al., 2017). Those increased wind speeds could have led to further transport of the coarser material from the source area, enabling these materials to reach our study area (Anderson et al., 1991).

From our data we see that the sandy layers were deposited in approximately 7000 yr (radiocarbon dates below and above these layers). As the sediments are rather coarse (115.3 µm mean grain size), a fluvial deposition is more likely than an aeolian deposition (Strauss et al., 2012). Moreover, the lack of organic material makes fluvial deposition the more plausible process. Thus, we think that periodic flooding events of Lena River tributaries near our study area are a more likely source for the sediments. The original Yedoma deposits of the Yukechi area were most likely formed by deposition of silty sediments and fine organic material during seasonal alluvial flooding. The climatic changes (Diekmann et al., 2017; Murton et al., 2017) and the resulting higher water availability during the deposition period of the sandy layers may have caused changes in fluvial patterns on the Abalakh Terrace. More water could cause higher flow velocities under warmer climatic conditions and therefore increased erosive power, leading to the formation of new flow channels (Reineck and Singh, 1980).

With a climatic backshift to colder conditions during MIS 2, water availability decreased and silty organic-bearing material was deposited by seasonal flooding again on top of the sandy layers. This likely led to lake initiation on top of the Yedoma deposits. The underlying ground began to thaw and subside, forming the Yukechi Alas basin. During this process, ice was lost from the sediment and the ground subsided by at least 9 m (height difference of 9 m between YED1 and Alas1 surface). Surface or lake water was able to percolate through the unfrozen sediments. This becomes visible by the homogeneous water isotope signal similar to the YED1 surface sample (Fig. 7). Under the unfrozen aquatic conditions in the sediment, microbial activity started, resulting in the decomposition of the already small amount of organic material (Cole et al., 2001). When the lake drained, the sediments started to refreeze both upward from the underlying permafrost and downward from the surface, leaving a talik in between (Fig. 5). The subsided

ground indicates that core unit A4 (Fig. 6) lies beneath the lowest unit of the Yedoma core, Y4 (Fig. 4), while units A1 to A3 shrank due to thawing from approximately 2200 to 1200 cm length. The presence of large ice wedges in the area supports this theory of ground subsidence during thaw as it hints to large excess ice contents of the ground (see Fig. S7) (Soloviev, 1959). These subsidence processes might represent the future path of the Yukechi Yedoma deposits, as already an initiating talik of approximately 150 cm was found at the YED1 site (Fig. 3a). This is caused by ground temperature warming which itself is affected by snow layer thickness and air temperatures and more, and could lead to Alas development (Ulrich et al., 2017a).

## 5.3 Central Yakutian Yedoma deposits in a circumarctic and regional context

Strauss et al. (2017) report a mean organic carbon density for the upper 3 m of Yedoma deposits in the Lena-Aldan interfluve of 25 to 33 kg/m$^3$ based on data of Romanovskii (1993) and Hugelius et al. (2014). Using a bootstrapping approach (Jongejans and Strauss, 2020) we found a much lower organic carbon density of $4.48 \pm 1.43$ kg/m$^3$ for the top 3 m of the YED1 core. For Alas1, an organic carbon density of $6.93 \pm 2.90$ kg/m$^3$ was calculated for the top 3 m. Taking the area covered by each deposit type within the Yukechi Alas landscape into account and the ice wedge volumes estimated by Ulrich et al. (2014) (see Section 2 and Fig. S2) into account, this results in a mean organic carbon density of only 4.40 kg/m$^3$ for the top 3 m of dry soil at the Yukechi study site. This landscape scale carbon stock density includes the entire study area (1.4 km²), including all water bodies (approximately 0.18 km²), which we assumed to contain no soil carbon. This means that both the average Yukechi site carbon density and our individual cores' carbon densities are substantially below the range (25 to 33 kg/m$^3$) reported by Strauss et al. (2017). This strong difference between previously published and our new data from the same region can only be explained by high depositional heterogeneity of the Central Yakutian permafrost landscapes that was not represented in the earlier dataset of Strauss et al. (2017) in sufficient detail. Geographically, the Yukechi area is located in one of the southernmost Yedoma areas in the Yedoma domain, which could be a reason for the differences to previously studied Arctic deposits (Schirrmeister et al., 2013; Strauss et al., 2013; Jongejans et al., 2018). Results of Siewert et al. (2015) for the Spasskaya Pad/Neleger site in a similar setting also differ greatly from our findings at the Yukechi site, showing carbon densities of approximately 19.3 kg/m$^3$ for the top two meters of larch forest-covered Yedoma deposits and approximately 21.9 kg/m$^3$ for the top two meters of grassland-covered Alas deposits in a setting similar to the Yukechi site.

In general, Yedoma deposits are estimated to hold 10 +7/-6 kg/m$^3$ for the whole column within the Pleistocene Yedoma deposits (approximate depth of 25 m) (Strauss et al., 2013). Jongejans et al. (2018) calculated a larger organic carbon stock of $15.3 \pm 1.6$ kg/m³ for Yedoma deposits found on the Baldwin Peninsula in Alaska. Another study by Shmelev et al. (2017) stated a Yedoma carbon stock of $14.0 \pm 23.5$ kg/m³ for a study region in northeastern Siberia between the Indigirka River and the Kolyma River.

Assessing the carbon inventory of the full-length Central Yakutian cores examined in this study, we estimated an organic carbon density of $5.27 \pm 1.42$ kg/m³ for the sediments of the YED1 core down to a depth of 22.12 m bs,

excluding the ice wedge. The organic carbon density within the Yukechi Yedoma is approximately two to three times lower than estimated in previous studies of deep Yedoma deposits (Strauss et al., 2013; Shmelev et al., 2017; Jongejans et al., 2018). Even when including roughly 10 m of organic carbon-free material, the higher carbon densities for the whole cores (compared to the carbon densities of the first 3 m) show that large portions of organic carbon are stored below 3 m. The Alas1 core contains slightly more organic carbon with a mean value of $6.07 \pm 1.80$ kg/m³ organic carbon for the whole core (19.72 m), which is about 20 % of the mean thermokarst deposit carbon content of 31 +23/-18 kg/m³ stated by Strauss et al. (2017). Within the Alas core, organic carbon storage is slightly higher in the top 3 m (approximately 14 % more than below). This likely is a result of former lake coverage that led to accumulation of organic-richer lake sediments found in the upper part of Alas1. Most likely there was enhanced growth of aquatic plants with at the same time reduced decomposition of the input organic material due to anaerobic conditions during the lake phase.

## 6 Conclusions

We conclude that low organic carbon contents encountered in sections of both cores are not caused by decomposition of originally high organic matter contents but rather are a legacy of the accumulation of organic-poor material during the late Pleistocene MIS 3 and MIS 2 periods. The most likely landscape scenario causing the differences in sediment and organic carbon characteristics during the Pleistocene deposition is the temporary existence of tributary rivers on the Abalakh Terrace with varying flow velocities and alternating paths as a result of climatic changes or local landscape dynamics. While with the onset of the Holocene the sedimentation on the Yedoma upland ceased, the Alas was affected by thaw, subsidence and lake formation processes, resulting in a compaction of sediments in situ as well as causing higher C inputs under lacustrine conditions in the upper parts of the sediments.

We further show that the Yedoma deposits at this site down to a depth of 22 m are characterized by rather low organic carbon contents, often less than 1 wt% TOC, resulting in a mean C density of only ~ 5 kg/m³.

Hence, the studied Yukechi Yedoma deposits store less carbon than other, comparable Yedoma Ice Complex deposits in the Central Yakutian area. However, there have been comparatively few studies on this so far. The biogeochemical impact of permafrost thawing in the Yukechi area might therefore be smaller than generally assumed for Yedoma deposits, as this area does not feature the high carbon stock estimates and high ice contents of other previously studied localities in Central Yakutia and elsewhere in the Arctic.

The permafrost characteristics found in the Alas core reveal that its composition and stratigraphy before lake formation and disappearance was very similar to the Yedoma core material. Its past development including thaw, the loss of old ice and surface subsidence, along with sediment compaction, shows a possible pathway for the Central Yakutian Yedoma deposits under the influence of global climate change.

## Data availability

The measurement data and laboratory results are available via PANGAEA at https://doi.org/10.1594/PANGAEA.898754 (Windirsch et al., 2019). A detailed core log is available for YED1 at https://doi.org/10.1594/PANGAEA.914874 and for Alas1 at https://doi.org/10.1594/PANGAEA.914876 (Windirsch et al., 2020b, a).

## Supplement link

## Author contribution

JS designed the study concept. TW conducted the laboratory work, analyzed the laboratory results, prepared the graphics, and led the writing of this paper. GG and AF led the drilling expedition in 2015. JS, MU, and PK participated in the drilling fieldwork. GG and JS supervised the data analyses and provided expertise on thermokarst processes and cryostratigraphy. LS provided expertise on grain-size characteristics and Central Yakutian permafrost genesis. MF designed the maps and provided expertise on Yedoma and thermokarst-affected carbon. LJ developed the bootstrapping routine and provided expertise on carbon stock upscaling. JW developed the age-depth models and worked on age calibration and contextualization. TO interpreted the water isotope results and provided context for the isotope data. JS took part in the laboratory work and provided expertise in permafrost carbon processes. All authors contributed to commenting and editing the manuscript.

## Competing interests

The authors declare no conflict of interest.

## Acknowledgements

This study is based on a joint field campaign of the ERC PETA-CARB project (Starting Grant #338335) and the DFG project UL426/1-1 and was carried out in cooperation with the Melnikov Permafrost Institute, Siberian Branch of Russian Academy of Sciences. TW was funded by the PoGS and LJ was funded by the DBU. The field campaign was supported by Avksentry P. Kondakov. We thank Dyke Scheidemann (Carbon and Nitrogen Lab [CarLa]) as well as Mikaela Weiner and Hanno Meyer (Stable Isotope Lab) from AWI for assistance in the laboratory. Planet data were provided freely trough Planet's Education and Research program. We thank Candace O'Connor for language correction.

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

**Table 1 - Radiocarbon measurement data and calibrated ages for YED1 and Alas1 bulk organic material samples.**

| Core | Mean sample depth [cm bs] | $^{14}$C age [yr BP] | +/- [yr] | F14C | +/- [%] | Calibrated ages (2 σ)* [cal yr BP] | Mean age [cal yr BP] | Core unit | AWI no. |
|------|---------------------------|----------------------|----------|------|---------|------------------------------------|----------------------|-----------|---------|
| YED1 | 157.5 | 18064 | 104 | 0.1055 | 0.83 | 21582–22221 | 21890 | Y1 | 1543.1.1 |
|      | 298 | 25973 | 88 | 0.0394 | 1.09 | 29822–30640 | 30268 | Y1 | 1544.1.1 |
|      | 589.5 | 35965 | 184 | 0.0114 | 2.29 | 40116–41118 | 40608 | Y1 | 1545.1.1 |
|      | 1636 | > 49000 | N/A | 0.0017 | 6.66 | N/A | N/A | Y3 | 1547.1.1 |
|      | 1998.5 | 45854 | 501 | 0.0033 | 6.23 | 48202–calib. limit | 49232 | Y4 | 1548.1.1 |
| Alas1 | 199 | 12826 | 57 | 0.2026 | 0.70 | 15144–15548 | 15287 | A1 | 1549.1.1 |
|      | 812.5 | 23615 | 151 | 0.0529 | 1.88 | 27478–27976 | 27729 | A2 | 1550.1.2 |
|      | 1530.5 | 42647 | 364 | 0.0049 | 4.53 | 45172–46619 | 45870 | A4 | 1551.1.1 |
|      | 1967.5 | 39027 | 251 | 0.0078 | 3.12 | 42478–43262 | 42865 | A4 | 1552.1.1 |

* calibrated using Calib 7.1 (Stuiver et al., 2018) equipped with IntCal 13 (Reimer et al., 2013)

**Table 2 – SOC contents for the individual core units, based on the bootstrapping results; calculations were done for 1 m²; the measurement data used in the bootstrapping approach (bulk density, TOC density) are provided in the data sheet in the PANGAEA repository; * refers to samples with TOC content < 0.1 wt%; for organic carbon pool calculations, we assumed a TOC of 0.05 wt% for these samples; note: we excluded unit Y2 in the calculations.**

| Core | Depth [cm bs] | Number of samples used in bootstrapping | Mean dry bulk density [$10^3$ kg/m³] | Mean TOC content [wt%] | Mean SOC content (bootstrapping results) [kg/m³] |
|------|---------------|------------------------------------------|--------------------------------------|------------------------|--------------------------------------------------|
| YED1 | 0 – 300 | 7 | 1190 | 0.42 | 4.48 ± 1.43 |
|      | 0 – 714 (unit Y1) | 13 | 1090 | 0.59 | 8.31 ± 1.41 |
|      | 1010 – 1927 (unit Y3) | 18 | 1172 | 0.10 | 0.86 ± 0.32 |
|      | 1927 – 2235 (unit Y4) | 5 | 910 | 1.14 | 11.50 ± 1.36 |
|      | total core | 36 | 1105 | 0.46 | 5.27 ± 1.42 |
| Alas1 | 0 – 300 | 5 | 1257 | 0.51 | 6.93 ± 2.90 |
|      | 0 – 349 (unit A1) | 6 | 1214 | 0.44 | 5.00 ± 2.55 |
|      | 349 – 925 (unit A2) | 6 | 998 | 0.05* | 0.50 ± 0 |
|      | 925 – 1210 (unit A3) | 4 | 1299 | 0.05* | 0.66 ± 0.01 |
|      | 1210 – 1980 (unit A4) | 12 | 1377 | 0.83 | 11.03 ± 1.62 |
|      | total core | 28 | 1250 | 0.47 | 6.07 ± 1.80 |

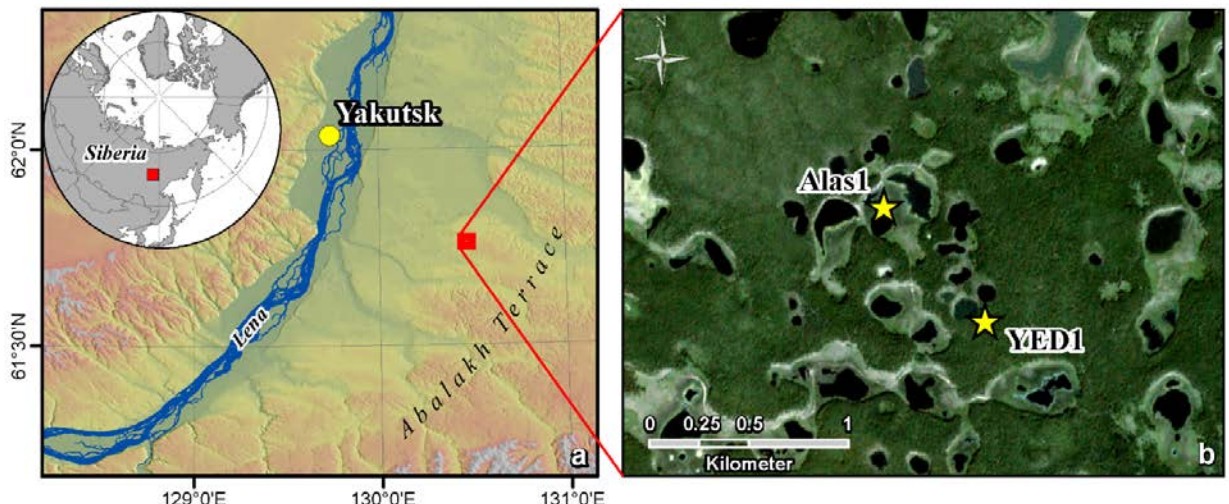

**Figure 1 – Study site overview; a: location of the Yukechi Alas study site in Central Yakutia on the edge of the Abalakh Terrace (Circumpolar digital elevation model, Santoro and Strozzi, 2012); b: locations of the Alas1 and the YED1 coring sites within the Yukechi Alas landscape (Planet OrthoTile, acquisition date: 7 July 2018; Planet Team (2017)).**

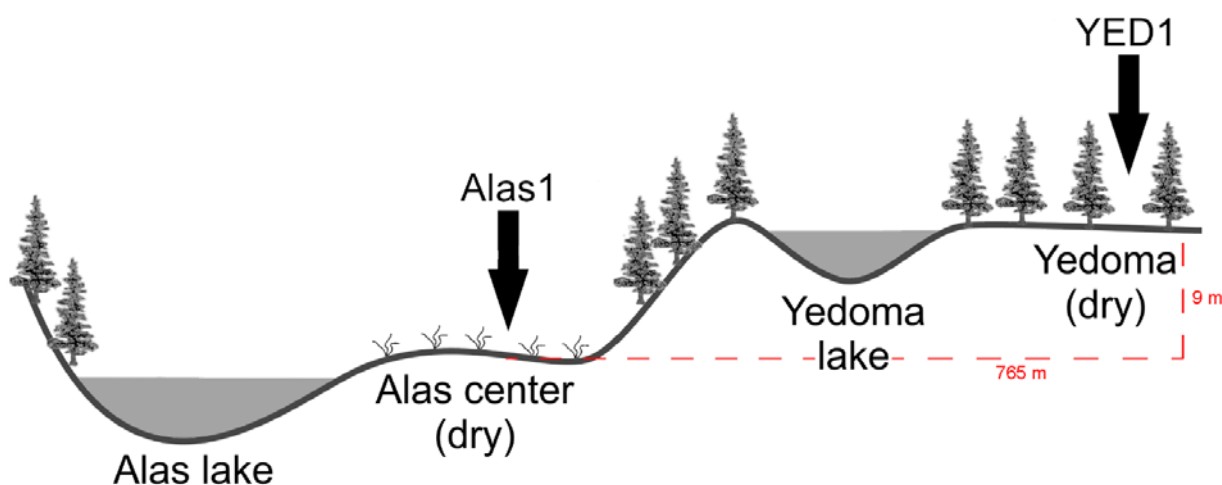

**Figure 2 – Setting of the drilling locations for the Alas1 and YED1 cores showing distance and height difference between the locations (vertical scale exaggerated); the terms "Alas lake" and "Yedoma lake" are chosen after Ulrich et al., 2017a in accordance to the deposit type in which the thermokarst lakes are located; following Crate et al., (2017), the Yedoma lake can also be called "dyede" due to its development stage.**

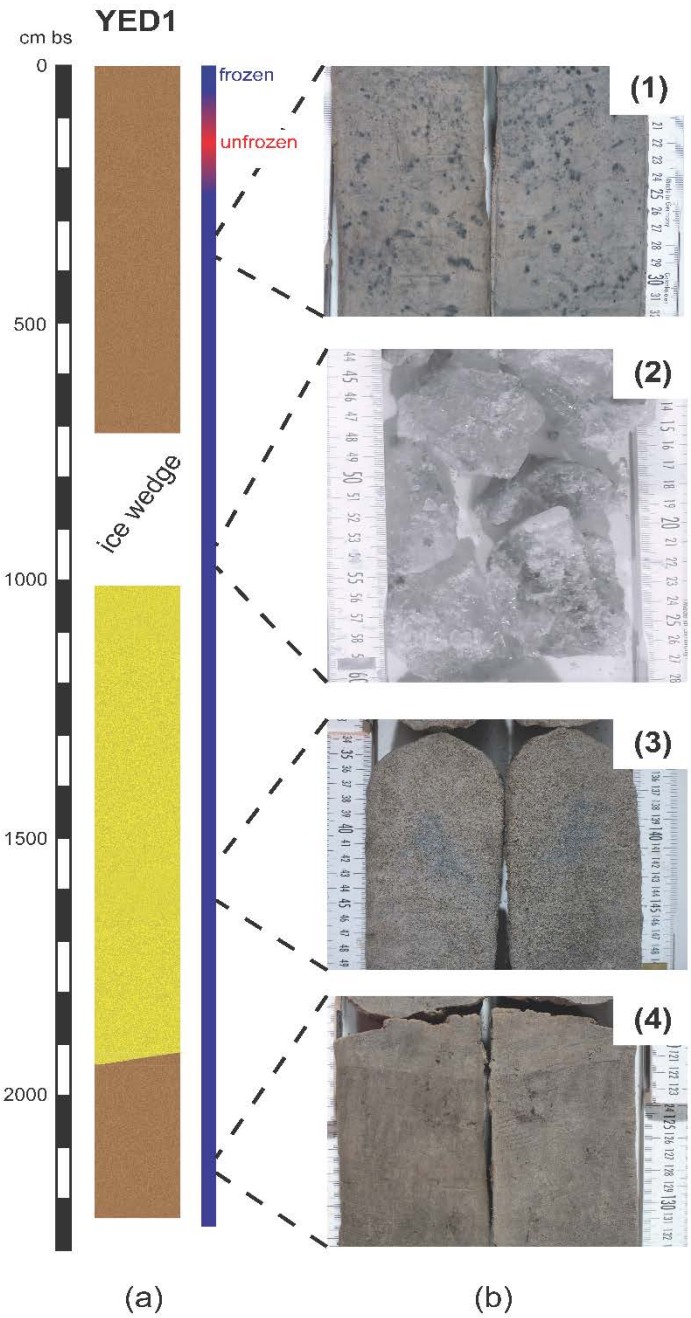

**Figure 3 – a:** overview of the Yedoma core; depth given in cm bs; state after core retrieval is given by colors: blue = frozen, red = unfrozen; location of the ice wedge is labelled; brown illustrates silty sediments, yellow represents sandy sediments; **b:** detailed pictures of the YED1 core; (1) 332–317 cm bs, picture of unit Y1 showing black organic-rich inclusions within the grey silty matrix; (2) 960–944 cm bs, picture of the wedge ice in Y2; (3) 1549–1532 cm bs, picture of Y3, showing the coarse sandy material with no visible cryostructures or organic material; (4) 2133–2117 cm bs, picture of Y4, showing the grey silty matrix with some dark organic dots.

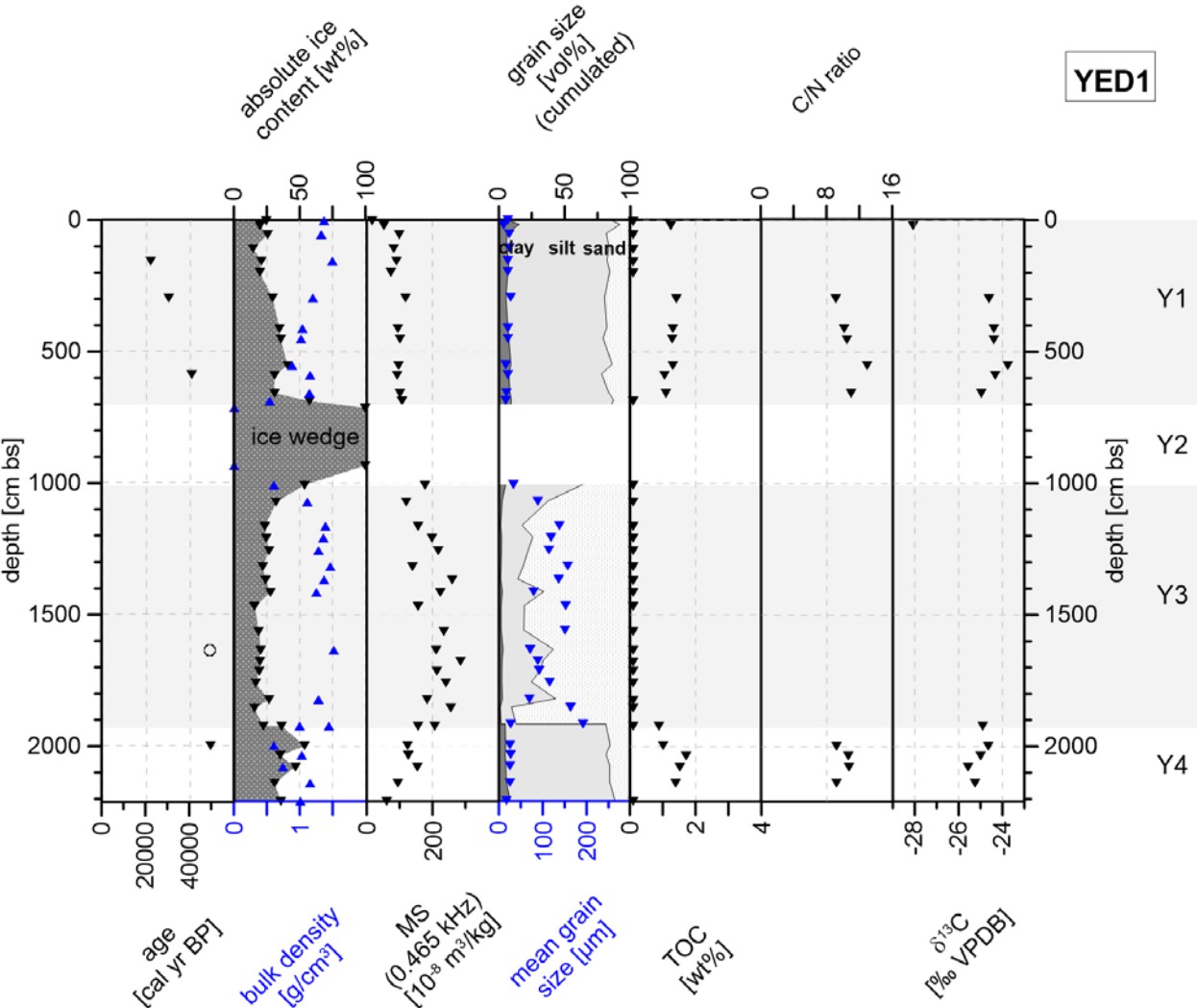

**Figure 4 – Characteristics of the Yedoma core YED1: radiocarbon ages, absolute ice content, bulk density, magnetic susceptibility (MS), grain size composition, mean grain size, total organic carbon (TOC) content, carbon-nitrogen (C/N) ratio and stable carbon isotope (δ¹³C) ratio; hollow circle indicates an infinite radiocarbon age; grey/white areas mark the different stratigraphic units (Y1 to Y4).**

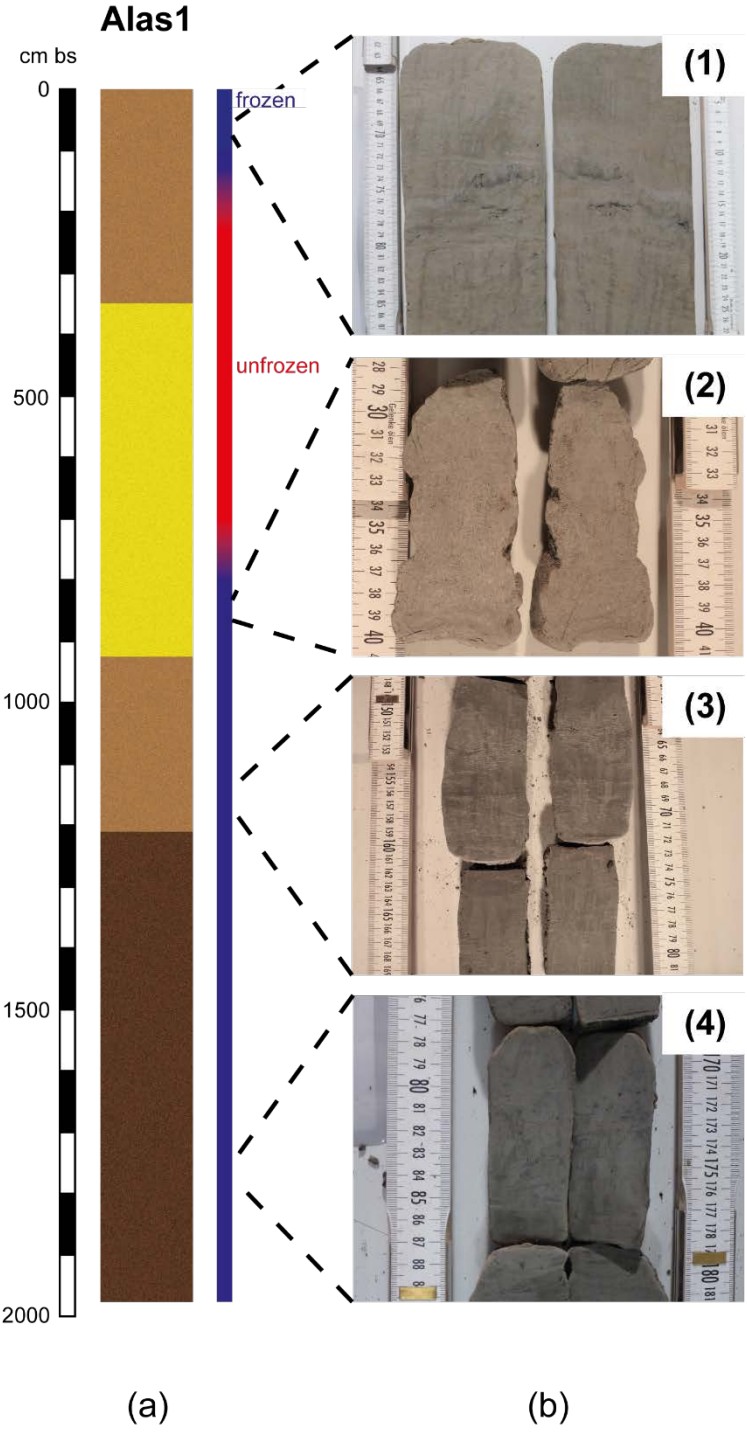

(a)  (b)

**Figure 5 – a: overview of the Alas1 core; depth given in cm bs; state after core retrieval is given by colors: blue = frozen, red = unfrozen; light brown marks silty material, yellow marks sandy material, dark brown marks silty material containing more organic material; b: detailed pictures of the Alas1 core; (1) 88–64 cm bs, picture of A1 showing the silty grey matrix**

including dark organic structures; (2) 840–828 cm bs, picture of the sandy A2 unit; (3) 1169–1148 cm bs, picture of A3 showing a silty grey matrix with some darker organic dots; (4) 1781–1767 cm bs picture of the fine-grained silt-dominated A4 unit including black organic-rich inclusions.

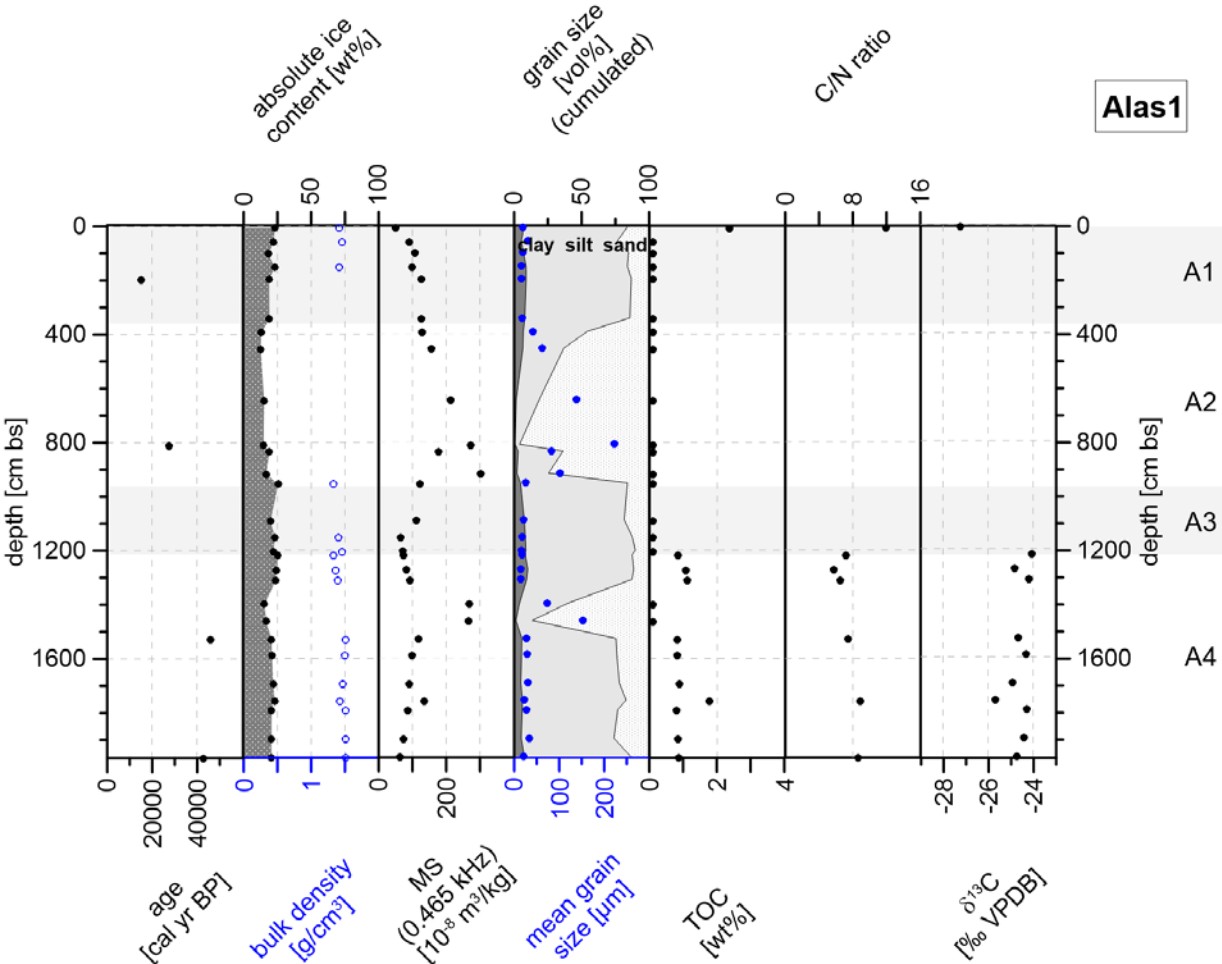

**Figure 6 – Characteristics of the Alas1 core: radiocarbon age, absolute ice content, bulk density, magnetic susceptibility (MS), grain size composition, mean grain size, total organic carbon (TOC) content, carbon-nitrogen (C/N) ratio and stable carbon isotope (δ13C) ratio; grey/white areas mark different stratigraphic units (A1 to A4).**

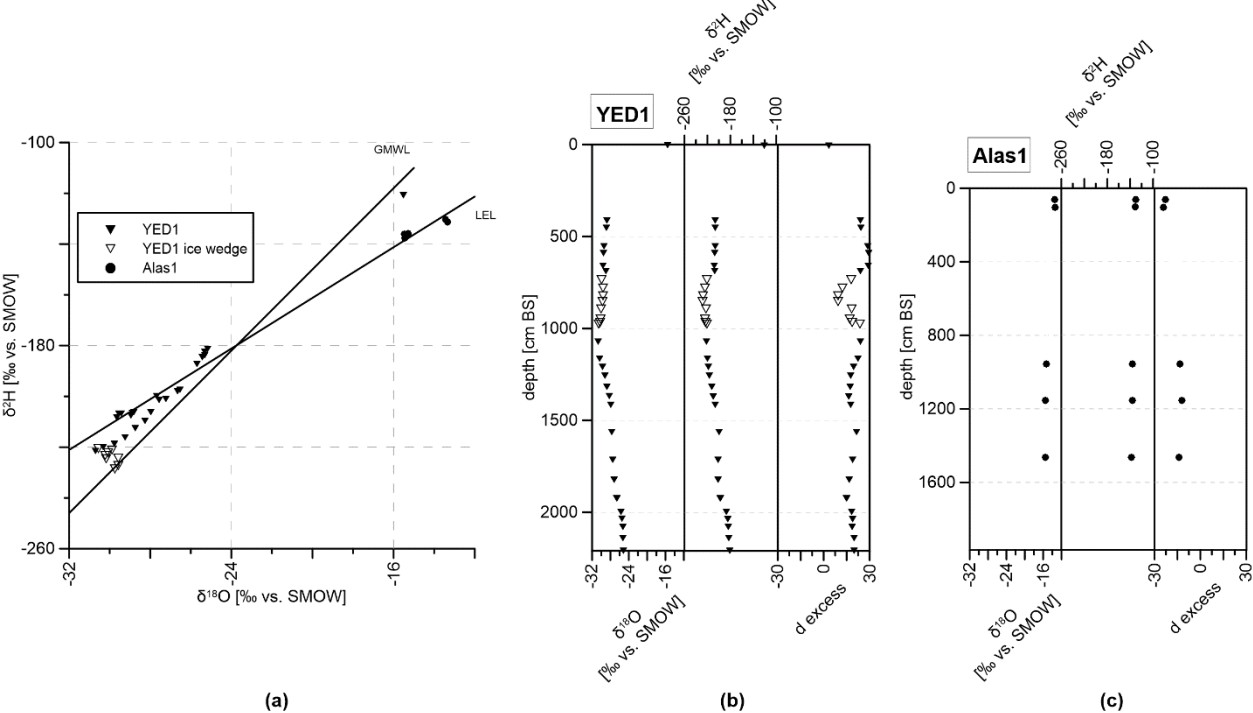

**Figure 7 – The characteristics of water stable isotopes in the studied sediment cores; a: stable hydrogen (δ²H) and oxygen (δ¹⁸O) isotope ratios of YED1 pore ice (black triangles), YED1 ice wedge ice (hollow triangles), and Alas1 pore ice and pore water (black dots) [‰ vs. SMOW]; global meteoric water line GMWL: δ²H=8*δ¹⁸O+10; local evaporation line LEL of Central Yakutia (based on data compiled until 2005 after Wetterich et al., 2008); b: oxygen isotopes, hydrogen isotopes and d excess values of YED1 plotted over depth; c: oxygen isotopes, hydrogen isotopes and d excess values of Alas1 plotted over depth.**