# Peer review of "Organic Carbon Characteristics in Ice-rich Permafrost in Alas and Yedoma Deposits, Central Yakutia, Siberia"

_Biogeosciences, 2019_

## Referee Comment (RC1) · Anonymous Referee #1 · 27 Jan 2020

The paper "Organic Carbon Characteristics in Ice-rich Permafrost in Alas and Yedoma Deposits, Central Yakutia, Siberia" by Windirsch et al. reports detailed analyses on carbon and ice contents, stable water isotopes, soil grain size distribution, and age estimates for two long ground profiles in the Central Yakutia, Russia. The presented materials are rare and highly valuable to understand landscape development and contents in the permafrost of the Eastern Siberia. Although I expect this paper to be finally published in Biogeosciece, the authors have not fully utilized, described, and discussed the data presented. The importance of this paper is the rareness of the sample core. I encourage the authors to enrich their descriptions about each core unit as a valuable drilling log of a permafrost region. Some portions in Discussion are

not logically constructed. For the main datasets of stable water isotopes and carbon parameters, a more in-depth and quantitative discussion is anticipated. Although the authors want to focus on the organic carbon characteristics as indicated in the title, more comprehensive discussion and interpretation must be done using available water geochemistry, cryostratigraphy, grain size distribution, magnetic susceptibility, etc. The usage of references or previous studies is poor and is often not clear or inappropriate. The discussion of sedimentation rates and their changes is problematic and too speculative. Radiocarbon age of bulk soil organic matter (SOM) usually much less reliable than that of macro plant remains. In permafrost environments, old (easily more than ten thousand years) carbon can co-exists in the same sedimentation stage, or younger carbon can be incorporated in a deeper layer. High mobility of SOM and geomorphological processes such as thermokarst and cryoturbation give a large uncertainty to sedimentation processes. Without rigorous consideration about the validity of the age model based on bulk SOM, the discussion about the changes in sedimentation rates does not make sense. As the authors explained, the upper part of the Alas1 accumulated by Yedoma deposit thaw. The Yedoma thaw involves differential ground subsidence and mixing of ground material through the thermokarst processes, which induce destruction of original stratification. The authors should discuss the limitations of the obtained dating data and build further discussion only based on reliable information. Below, I listed individual points to be revised or to be clearly explained in the revised manuscript to achieve a publication quality.

P1-L25-26: What is "a potential theory of Holocene influence"? P1-L30: What do you mean ". . . the Yedoma core can be duplicated."? P1-L31-36: The authors describe using numerous "different" and "differ" works, but I suggest to revise to explain them more concretely. I could not understand how different. L36: How the Alas core gives clear insights? P2-L16: Why Yedoma deposits highly vulnerable to thaw? Usually, high-ice content permafrost is more robust to thaw because of larger latent heat storage than drier permafrost. We can say Yedoma is highly susceptible to thermokarst though. P2-L24: "That these processes. . ." ??? P3-L25: active layer thickness should

be larger/smaller rather than higher/lower. P4-L24: What do you mean by "...if 'previously' unfrozen,..."? What is "artificial roots (Rhizones)"? Please describe more details about the water extraction and explain how you avoid evaporation from the sample water. P4-L26: Please add information about subsample intervals. P4-L28: Do you have a reason to use "absolute" ice content? I think gravimetric ice content makes clearer what you are dealing with. P4-L29: I suggest to distinguish liquid and solid water more clearly. There needs more explanation to "water which froze after drilling." P5-3.2.5: Please describe the accuracy of the stable isotope analyses. P6-3.2.6: Please explain briefly about the bootstrapping approach. P7-L21: What do you mean by "...dated with an infinite radiocarbon sample."? P7-L26: Please describe more about ice characteristics in this unit. Add some photos in the manuscript. P8-L26: If the active layer thickness reaches to 200 cm, the seasonal thaw depth touches the top of the talik (160cm). Does this mean the thickness of the seasonally freezing layer is 160cm? Section 4.3: Why some YED1 points plot above the GMWL on the LEL? Could you add some explanation in Discussion? Which portion of the core are they? P10-L27-29: Similarities between which units of them? P11-L2-5: Please rewrite this sentence. It is hard to understand probably because of the usage of too many conjunctions. P11-L7-8: I see varying values of d13C with depth in the profiles. What does your "constant" mean? You are comparing one data point in the active layer and data from other depth? I think it is not valid to compare decomposition rates between organic carbon currently decomposing in the active layer and ones formerly exposed to decomposition for an unknown period then frozen. As far as I see the profile, both C/N ratio and d13C fluctuate with depth even in the same unit. Is this reflected by varying decomposition rates? P11-L16-20: "The similarity of the low C/N ratios from both cores..." Which both cores? Your YED1 and Alas1? Or are you comparing with other Yedoma sediments found near the Arctic sea? I could not understand why this similarity supports the sedimentation of organic-poor materials. Physical conditions (such as climate and hydrology) between the comparing regions are quite different.

P11-L22-: Please show both profiles in one graph for better comparison. The heights

of 0 m of the graphs of d18O profiles differ in Fig. 7 b/c, but not 9 m illustrated in Fig. 2. You can reflect the relative height difference in the same graph if you want. Similarly, please add a graph showing d-excess profiles of both cores in Fig. 7 (not in supplemental figures) and discuss evaporation and freeze-out fractionation processes that might have affected to your profiles. I think Fig. S7 should be shown as the main figure, probably, combining with Fig. 7 b/c and d-excess profiles if you want to discuss dD profiles. P11-L21-26: This entire paragraph is not logically described and I do not find sounding discussion. Why the age inversion and stable water isotope signals are related? Why particularly 35 K BP is the timing of the shift in sediment input? "This would have been . . ." What "This" indicates here? P11-L27: "permanently" means always and forever. You should assign a more specified period of time when you discuss topics like in your manuscript. It is impossible to be in a permanently frozen state throughout the depositional history. P11-L28: Please clarify depths for "the lower Yedoma pore ice" and "the uppermost sample." "the uppermost sample" is the data at 0m in YED1? P11-L31: "The very similar.." to what? P12-L1: The original data were from Popp et al. (2006)? P12-L2: "offset" of what? Your ice-wedge values against what? P12-L4: As I mentioned above, the age constraints around 40-50 K BP need reconsideration. Furthermore, isotope signals of pore water do not simply show climate change. Lots of complicated fractionations and changes in seasonal meteoric water components could affect the signals. P12-L5-: This topic sentence doesn't make sense to me from your discussion above. What do you mean "lake converge"? Your discussion mixes decomposition rates during Yedoma deposition and during alas formation. P12-L25-: Cryostructure of frozen sediments mainly depends on soil grain size distribution, freezing rate, and moisture content. Sandy sediments usually have fewer pore spaces comparing to sediments consist of finer soil particles. Sandy sediments tend to have structureless cryostructure because sandy particles have a limited amount of unfrozen water under cryotic conditions. Section 5.2 the first paragraph: I think the authors are misunderstanding or misinterpreting the papers Wetterich et al. (2009), French and Shur (2010), and Iwahana et al. (2014) as to the mechanism of cryostructure formation and sediment/carbon movements in the ground. Section 5.2 the second paragraph: Discussion in this paragraph makes sense to me. This discussion should be placed in Section 5.1 as it is tightly related to carbon accumulation and loss at this site. P13-L23: Why "organic-bearing"? I thought you are discussing the transportation of organic-poor sediments. Could you discuss this sandy deposit could be Aeolian or not using your grain size data? P13-L25: Where the "7000 yr" came from? Aeolian materials cannot be organic-poor? P13-L29: What do you want to refer to Diekmann et al. (2017)? The climate changes? P13-L4: What the "this" indicate? The deposition of silty organic-bearing material on top of the sandy layers, decreased water availability, or a climatic backshift to colder conditions? In any case, I could not understand the logic. P13-L5: Why you refer to Grosse et al. (2013) for "The underlying ground began to thaw"?? P13-L7: "as visible in the . . ." should be explained concretely. P13-L12: "shrunk . . . from . . . 2200 to 1200cm.." Please discuss the possibility of subsidence using excess ice volume information. Is there enough excess ice content in the initial Yedoma to reduce its volume from 2200 to 1200cm thick? P13-L12-13: Again, what "this" indicates here? P13-L15-16: I don't understand why you started to discuss the tipping point suddenly. Is this relevant and supported by your results? Section 5.3 the first paragraph: How the 4.40 kg/m3 was derived? Please explain the relation to 4.48 and 6.93 for the two sites? This could be the first carbon stock measurement below 3m for two sites in the Central Yakutian Yedoma in English literature, but have you checked Russian literature? Please discuss how representative your two cores for the Central Yakutian Yedoma. P16-L1-2: I could not understand what you wanted to indicate from this sentence. What do you mean by "drainage"? How did the alas core reveal its composition and stratigraphy before lake formation? Fig. 2: Yedoma lake should be thermokarst lake? Or by local name, "dyuyodya" (Soloviev, 1973) or "dyede" (Crate et al., 2017)? Soloviev, P. A. (1973), Thermokarst phenomena and landforms due to frost heaving in Central Yakutia, Biuletyn Peryglacjalny 23, 135-155. Crate, S., et al. (2017), Permafrost livelihoods: A transdisciplinary review and analysis of thermokarst-based systems of indigenous land use, Anthropocene, 18, 89-104,

doi: https://doi.org/10.1016/j.ancene.2017.06.001. Fig. 3: Include photos of each unit including ice-wedge layer. Put "YED1" in the figure. Fig. 5: Please use the same white balance and adjust the exposure/brightness of the core photos as in Fig. 3. Include core photos of the rest two units too. Put "Alas1" in the figure. Fig. 4 & 6: Please used different markers for absolute ice content and bulk density so that people without color printer environment or color-blind can easily distinguish the two. Large caption YED1 and Alas1 within the figures would be helpful. Fig. 7 & S7: Combine Fig 7 b/c and S7 and add a graph of d-excess. Please use open triangle markers for ice-wedge points. The half-open triangle hinders instant discrimination between markers. Add labels "YED1" or "Alas1" on top of each graph. Fig. S2: The blue markers with age uncertainties are not discernible. Please revise them for a clearer presentation. I do not think the information on the model confidence by black shades is meaningful and necessary in your radiocarbon age of soil organic matter. For all graphs with depth axis: Please use the meter scale, not cm.
* * *

---

## Referee Comment (RC2) · Anonymous Referee #2 · 16 Feb 2020

The manuscript by Windirsch et al. "Organic Carbon Characteristics in Ice-rich Permafrost in Alas and Yedoma Deposits, Central Yakutia, Siberia" examined and compared the organic carbon storage and characteristics from two rare and deep cores using different methods (C%, soil texture, 14C age, ice content, etc..). Since such deep cores are very rare, this study is very important and gives us valuable inside information about the history of these deep deposits. The scientific question and the used methods are well established, however, there is still room for some improvement. Overall, the discussion section is rather weak and speculative, and the based conclusions too shaky. One reason, most of the used methods were nor fully incorporated in the discussion section. Some results as for example 14C inversions or magnetic susceptibility are not really discussed and explained. Nitrogen data not presented, even though obviously available. Conclusion to short, missing main points as for example that the global estimate of SOC in Yedoma might be by far overestimated and possibly not so vulnerable due to rather low ice contents. Also, recent publications suggest Yedoma being not extremely vulnerable. Missing older Russian Literature. Below I list more specific points, which should be addressed before final publication. P.1 L. 15: "has not yet", actually parts of Yedoma are already in the active carbon cycle; P1 L: 30 "very carbon poor", please add number; P2 L2 "provide"???, and why is it important; P2 L6, reference on co2 in the atmosphere not appropriate. Use actual data source; P2 L16 "vulnerable" recent paper by Kuhry et al. states that Yedoma is rather stable. ; P2 L20 Is this the case only for carbon stored within Yedoma deposits???; P2 L21 released how? I guess you mean as carbon dioxide.; P2 L25, please add ref.; P2 L27 Explain why below 3m less understood? P2 L29 missing space; P2 L32 "822 Gt" this is wrong, the number refers not only to Yedoma, but refers to combined permafrost SOC in soils (0–3 m), deltaic alluvium and Yedoma region sediments. P3 L5 Your 'only' 2 research questions are going under in the paragraph. Please make them stand out. P3 L24 "in Siberia", is there no data for MAAT at least for Central Yakutia? P2 L27, No more data or drilling campaigns since Soloviev, 1973? P4 L23 "approximately every 50cm? Why this distance? Why not using visual changes in the core for the increments? P6 L4 check paragraph spacing. P6 Since the main question is carbon, a chapter on how carbon estimates were calculated would be more then appropriate. P7 L18 "material composition???" soil texture. P9 L6 please remove rather, enough with the given SD. P9, when you compare the grain sizes, you change between median and mean, why? P10 L1 I don't fully understand the SOC calculations. How many samples were used for boot strapping? Please add a table comparing these two cores with SOC data, TN data, DBD data, to 3 m meter, to the different used units and to the total. P11 L1-5. And what exactly are you arguing? Also, a lot of speculations. Please stick to the point. Also, the rather obvious reason for the low C comes fist on page 13! P11 L13-15 But then the frozen sections should have the same C/N ratio, but they don't?

P11 L18 "decomposed" if, then rather pre-decomposed. P11 L21-26 Ok, so what is the reason for inversions? Section by far too short. And no discussion on the other dates. 11 L32 "indicate colder climate conditions" How, and what does it mean for the core? Any refs for that? P12 L2 "explained by climatic variations" Is this an assumption or is there data? Need refs for this statement. P13 The final reason for the low C% should me mentioned earlier. Also, this page can be shortened. P15 L15 & 23, repetitive sentence. Also, main conclusion, Yedoma SOC estimates likely too high. P15 L28. "high ice content.... vulnerable" actually the opposite. You showed very little ice content except one one ice wedge. Yedoma "had an estimated" ice wedge and lenses content of up to 90%. These two cores have far less. So these Yedoma deposits are not very vulnerable.

---

## Author Comment (AC2) · 4 Apr 2020

Reply to: Anonymous Referee #2, received and published: 16 February 2020

The manuscript by Windirsch et al. "Organic Carbon Characteristics in Ice-rich Permafrost in Alas and Yedoma Deposits, Central Yakutia, Siberia" examined and compared the organic carbon storage and characteristics from two rare and deep cores using different methods (C%, soil texture, 14C age, ice content, etc..). Since such deep cores are very rare, this study is very important and gives us valuable inside information about the history of these deep deposits.

[Figure]

Thank you very much for this general statement of importance of our study.

The scientific question and the used methods are well established, however, there is still room for some improvement. Overall, the discussion section is rather weak and speculative, and the based conclusions too shaky. One reason, most of the used methods were nor fully incorporated in the discussion section. Some results as for example 14C inversions or magnetic susceptibility are not really discussed and explained.

We are grateful for the review and acknowledge the reviewer's comments, which really improve and strengthened the paper. As suggested by the reviewer we extended and clarified the discussion and strengthened the conclusion. Please see our comments below.

Nitrogen data not presented, even though obviously available.

They have not been shown as most values did not reach detection limit. This has now been referenced and implemented in the result sections 4.1 and 4.2. Moreover, the data presented in the manuscript are available on the PANGAEA repository.

Conclusion to short, missing main points as for example that the global estimate of SOC in Yedoma might be by far overestimated and possibly not so vulnerable due to rather low ice contents

Yedoma ice contents are low in this specific case, which is described in the conclusions. With one site we can not revise circumarctic estimates. Therefore, general Yedoma vulnerability is not the topic of this discussion. As the drained thermokarst basins are 9 m deeper there is a substantial potential for Yedoma for severe changes. Using this study to revise studies basing on a Yedoma wide study site set (Strauss et al 2017, Schirrmeister et al 2011) is not appropriate to our mind. But nevertheless, as we state we make this point in the discussion but this is not one of the main conclusions.

Also, recent publications suggest Yedoma being not extremely vulnerable. Missing older Russian Literature

We checked on international available and peer-reviewed Russian literature and added some more suitable literature. We avoid citing non-peer-reviewed literature and literature that is not available to an international scientific audience. We would be happy to include more literature. Please use the comment function to add specific literature. Besides this, we concluded that Yedoma is highly vulnerable to thaw induced landscape changes due to loss of excess ice. However, as stated by Kuhry et al, the carbon stored in Yedoma deposits might not be highly vulnerable. We now implemented Kuhry et al. (2020) to the paper which was not published at the first submission of our paper.

Below I list more specific points, which should be addressed before final publication. P.1 L. 15:"has not yet", actually parts of Yedoma are already in the active carbon cycle;

We changed this to "this carbon becomes available to the recent carbon cycle.", as some of it is still frozen and immobilized.

P1 L: 30"very carbon poor", please add number;

Thank you, changed accordingly: We added "(< 0.1 wt%)" to the description, which is the detection limit for measuring the total organic carbon content.

P2 L2 "provide"???, and why is it important;

We changed this to "Permafrost deposits present one of the largest terrestrial carbon reservoirs."

P2 L6, reference on co2 in the atmosphere not appropriate. Use actual data source;

We changed the reference to 868 Gt which was calculated using the CO2 concentration of 407 ppm, measured in 2018. This was calculated using the conversion formula given by Friedlingstein et al (2019). Global carbon budget using a factor of 2.124 for conversion between ppm CO2 and Gt carbon after Ballantyne et al (2012) for well-mixed atmospheric conditions. The text now reads: "The estimated amount of frozen and unfrozen carbon stored in the terrestrial permafrost region is 1330 to 1580 gigatons (Gt) (Hugelius et al., 2014;Schuur et al., 2015), which is up to 45% more than what is

currently present in the atmosphere ($\sim$ 864 Gt, based on 407 ppm CO2 measured in 2018) (Ballantyne et al., 2012; Friedlingstein et al., 2019)."

P2 L16 "vulnerable" recent paper by Kuhry et al. states that Yedoma is rather stable.;

Concerning landscape vulnerability: the general high ice content in Yedoma deposits makes them vulnerable to dramatic landscape changes caused by thaw under climate warming; Concerning C vulnerability: Kuhry et al (2020) does not state the opposite but concludes that thaw vulnerability does not mean a high vulnerability/lability of SOM. This sentence was changed as follows: "In the context of global climate change, such high ice content with pore ice and syngenetic ice wedges render Yedoma deposits highly susceptible to thaw induced landscape changes (Schirrmeister et al., 2013) and ground volume loss causing surface subsidence."

P2 L20 Is this the case only for carbon stored within Yedoma deposits???;

Thank you, we changed "Yedoma deposits" to "permafrost".

P2 L21 released how? I guess you mean as carbon dioxide.;

We specified this by adding "... in form of gases such as carbon dioxide or methane,..."

P2 L25, please add ref.;

We added "(Strauss et al., 2013)" to the corresponding sentence.

P2 L27 Explain why below 3m less understood?

This is due to the fact that only very few studies have been examining long permafrost cores. We clarified that in the text: "...especially below 3 m, are still poorly studied, as only few studies examining long Siberian permafrost cores have been conducted (Zimov et al., 2006;Strauss et al., 2013;Shmelev et al., 2017)."

P2 L29 missing space;

changed as suggested

P2 L32 "822 Gt" this is wrong, the number refers not only to Yedoma, but refers to combined permafrost SOC in soils (0–3 m), deltaic alluvium and Yedoma region sediments.

Thank you for notifying. We changed it to "permafrost region".

P3 L5 Your 'only' 2 research questions are going under in the paragraph. Please make them stand out.

We made the research questions more visible by adjusting the paragraphs.

P3 L24 "in Siberia", is there no data for MAAT at least for Central Yakutia?

Thank you for your suggestion. We changed this to "The contemporary mean annual air temperature in Central Yakutia (measured at Yakutsk Meteorological Station) is -9.7 °C."

P3 L27, No more data or drilling campaigns since Soloviev, 1973?

There are actually few accessible studies on the thickness of the Yedoma in Central Yakutia. The most cited studies are indeed Soloviev, 1959 and 1973. We changed this statement and include both references as follows: "The Yedoma deposits in this region can be more than 30 m in thickness as was already shown by older Russian works (Soloviev, 1959, 1973)."

P4 L23 "approximately every 50cm? Why this distance? Why not using visual changes in the core for the increments?

The cores were indeed sub-sampled after visible stratigraphic changes. However, larger homogeneous sections were sampled approximately every 50 cm. We specified our subsampling approach as follows "Subsamples were equally distributed along the cores. According to visual changes, we covered all visible stratigraphic layers and we sampled at least every 50 cm in order to capture specific sediment properties."

P6 L4 check paragraph spacing.

Thank you, we adjusted it.

P6 Since the main question is carbon, a chapter on how carbon estimates were calculated would be more then appropriate.

Thank you. The methods of our carbon budget calculation are explained in chapter 3.2.6. We renamed the chapter title to "Statistics and bootstrapping approach for carbon budget estimations" to make this clearer.

P7L18 "material composition???" soil texture.

Changed accordingly

P9 L6 please remove rather, enough with the given SD.

Changed accordingly

P9, when you compare the grain sizes, you change between median and mean, why?

We corrected this and now use the mean for all data.

P10 L1 I don't fully understand the SOC calculations. How many samples were used for boot strapping? Please add a table comparing these two cores with SOC data, TN data, DBD data, to 3 m meter, to the different used units and to the total.

Thank you. We expanded the explanation on the bootstrapping approach. Also, we added a table to show SOC contents for the individual core units, based on the bootstrapping results. As TN data are not used in the bootstrapping approach, we did not include them in table 2 (please see below). These data are shown in the PANGAEA repository as most values are below the detection limit. Please see our comment further above.

P11L1-5. And what exactly are you arguing? Also, a lot of speculations. Please stick to the point. Also, the rather obvious reason for the low C comes first on page 13!

This was a misuse of "arguing", we apologize. We changed the sentence into: "On

the one hand, low TOC content could be a result of high organic matter decomposition during accumulation or during a thawed state, especially in thermokarst deposits. On the other hand, this could reflect low carbon inputs. The low stable carbon isotope data of our cores (between -24.06 and -27.24 ‰ are comparable to other studied sites from the Yedoma domain (Schirrmeister et al., 2013;Strauss et al., 2013;Jongejans et al., 2018), and suggest that the source signal was more or less constant with time.".". The reason for low C is discussed in more detail later in chapter 5.2.

P11L13-15 But then the frozen sections should have the same C/N ratio, but they don't?

They do not have the same C/N ratio, as the Alas1 core was once thawed completely, as far as we conclude from the water isotope data. To highlight this we change this part as follows: "In comparison to the mean C/N ratio of 10 in YED1, the mean C/N ratio of 8 for Alas1 may indicate that the Alas deposits were only slightly more affected by decomposition due to a temporary thawed state. The development of a talik in Alas1 probably led to increased decomposition of these sediment sections and thus leading to a reduced C/N ratio. As the carbon was freeze-locked in the YED1 core and therefore was not decomposed since deposition, this similarity in mean C/N ratios indicates that the original carbon state for the deposits at both coring sites were similar, assuming the same carbon source for both deposits."

P11 L18 "decomposed" if, then rather pre-decomposed.

Changed accordingly

P11 L21-26 Ok, so what is the reason for inversions? Section by far too short. And no discussion on the other dates.

We restructured this paragraph and clarified. We also added further discussion in the paragraph that now reads as follows: "We found age inversions in both cores with similar age and depth (YED1 49,232 cal yr BP, 1998.5 cm bs; Alas1 42,865 cal yr BP, 1967.5 cm bs) (Fig. 4 and 6, Fig. S2). While cryoturbation might seem an obvious

explanation, we suggest that this process did not play a major role here due to the long-term frozen state of YED1. Rather, we assume that the age inversions indicate a temporary shift in sediment input at approximately 35,000 cal yr BP causing some deposit reworking in the watershed and the incorporation of older material into younger sediments. In addition, the dating of bulk sediments very close to the maximum datable age of $\sim$ 50,000 yr BP may cause a high uncertainty in the absolute ages of sediment layers (Reimer et al., 2013). Therefore, the rather small age inversions (> 49.000 cal yr BP to 49,232 cal yr BP in YED1, and 45,870 cal yr BP to 42,865 cal yr BP in Alas1) could be a result of material mixture in dated bulk samples. The radiocarbon ages above this age inversion align well with a simulated sedimentation rate, as shown in figure S2."

P11 L32 "indicate colder climate conditions" How, and what does it mean for the core? Any refs for that?

We deleted this sentence, as it was unnecessary information without important implications for the core.

P12 L2 "explained by climatic variations" Is this an assumption or is there data? Need refs for this statement.

We restructured the whole paragraph and added more detail on isotope data and d excess: "The stable isotope ratio values of ice wedges (mean $\delta18O$ of -30 ‰ mean $\delta2H$ of -224 ‰ reflect winter precipitation and fit well into the regional pattern for MIS 3 ice wedges in Central and Interior Yakutia (Popp et al., 2006;Opel et at., 2019) while the d excess shows a much elevated value (16 ‰ compared to the regional pattern (Popp et al., 2006;Opel et al., 2019). It should be noted that the d excess values from the central core parts correspond well to the regional values from Mamontova Gora, Tanda and Batagay (Opel et at., 2019), while the others fit to those of the host sediments and are potentially overprinted by exchange processes between wedge ice and pore ice (Meyer et al., 2010). Due to the low number of datapoints, no meaningful co-isotopic

regression could be calculated. The stable isotope composition of pore ice shows a co-isotopic regression of $\delta 2H = 6.61\ \delta 18O - 18.0$ ($R^2 = 0.97$, n = 23), which is typical for Yedoma intrasedimental ice (Wetterich et al., 2011;2014;2016). The isotope values plot well above the GMWL of the cold season (Papina et al., 2017) suggesting a substantial proportion of (early) winter precipitation (usually characterized by high d excess values) for the pore ice, which is also evident for some units of the Batagay megaslump (Opel et al., 2019). The decreasing upward trend of pore ice isotopic $\delta$ values might point to a general cooling in Central Yakutia during the covered time span of our study. However, as it is accompanied by an opposite increasing trend in d excess, these values may be overprinted by secondary freeze-thaw processes in the active layer and rather reflect the strength of these fractionation processes (Wetterich et al., 2014)." Moreover, we strengthened our stable water isotope expertise by inviting Thomas Opel to the author team.

P13 The final reason for the low C% should be mentioned earlier. Also, this page can be shortened.

Thank you for this comment. We agree and moved this interpretation into section 5.1. We also restructured and shortened this chapter.

P15 L15 & 23, repetitive sentence. Also, main conclusion, Yedoma SOC estimates likely too high.

We removed the repetition. Thanks for this comment but following Schneider von Deimling et al (2015) the SOC estimations for Yedoma could still be correct, giving the Yukechi site as a low-carbon example. Also, due to the sediment diversity within the YED1 and Alas1 cores, the Yukechi Alas landscape does not represent a typical Yedoma composition.

P15 L28. "high ice content.... vulnerable" actually the opposite. You showed very little ice content except one one ice wedge. Yedoma "had an estimated" ice wedge and lenses content of up to 90%. These two cores have far less. So these Yedoma

deposits are not very vulnerable.

The high volumetric ice content in Yedoma deposits is mainly due to the presence of large ice wedges in the area (please see new figure S7 in the updated supplement), as in this case in our Yedoma core. We have chosen the Yedoma coring location in the centre of a polygon for not hitting an ice wedge. In this context, the Alas core was not included, as it represents already reworked Yedoma deposits in which the Pleistocene ice has largely melted. The pore ice content of the Central Yakutian Yedoma sediment deposits is comparatively smaller, but the sensitivity of the sediments to thawing processes is retained as for the ice wedges. This is also shown by the numerous large thermokarst landscape features in the region, but also by the currently observed very rapid thawing processes.

Please also note the supplement to this comment:
https://www.biogeosciences-discuss.net/bg-2019-470/bg-2019-470-AC2-supplement.pdf

———————————————————

**Table 2 – SOC contents for the individual core units, based on the bootstrapping results; calculations were done for 1 m²; the measurement data used in the bootstrapping approach (bulk density, TOC density) are provided in the data sheet in the PANGAEA repository; \* refers to samples with TOC content < 0.1 wt%; for organic carbon pool calculations, we assumed a TOC of 0.05 wt% for these samples; note: we excluded unit Y2 in the calculations.**

| Core | Depth [cm bs] | Number of samples used in bootstrapping | Mean dry bulk density [10³ kg/m³] | Mean TOC content [wt%] | Mean SOC content (bootstrapping results) [kg/m³] |
|---|---|---|---|---|---|
| YED1 | 0 – 300 | 7 | 1190 | 0.42 | 4.48 ± 1.43 |
| | 0 – 714 (unit Y1) | 13 | 1090 | 0.59 | 8.31 ± 1.41 |
| | 1010 – 1927 (unit Y3) | 18 | 1172 | 0.10 | 0.86 ± 0.32 |
| | 1927 – 2235 (unit Y4) | 5 | 910 | 1.14 | 11.50 ± 1.36 |
| | total core | 36 | 1105 | 0.46 | 5.27 ± 1.42 |
| Alas1 | 0 – 300 | 5 | 1257 | 0.51 | 6.93 ± 2.90 |
| | 0 – 349 (unit A1) | 6 | 1214 | 0.44 | 5.00 ± 2.55 |
| | 349 – 925 (unit A2) | 6 | 998 | 0.05* | 0.50 ± 0 |
| | 925 – 1210 (unit A3) | 4 | 1299 | 0.05* | 0.66 ± 0.01 |
| | 1210 – 1980 (unit A4) | 12 | 1377 | 0.83 | 11.03 ± 1.62 |
| | total core | 28 | 1250 | 0.47 | 6.07 ± 1.80 |

27

**Fig. 1.** Table 2

**Supplement:**

*Supplement of*

**Organic Carbon Characteristics in Ice-rich Permafrost in Alas and Yedoma Deposits, Central Yakutia, Siberia**

Torben Windirsch[1,2], Guido Grosse[1,2], Mathias Ulrich[3], Lutz Schirrmeister[1], Alexander N. Fedorov[4,5], Pavel Ya. Konstantinov[4], Matthias Fuchs[1], Loeka L. Jongejans[1,2], Juliane Wolter[1], Thomas Opel[1], and Jens Strauss[1]

[1]Alfred Wegener Institute Helmholtz Centre for Polar and Marine Research, Telegrafenberg A45, 14473 Potsdam, Germany
[2]University of Potsdam, Institute of Geosciences, Karl-Liebknecht-Straße 24-25, 14476 Potsdam, Germany
[3]Leipzig University, Institute for Geography, Johannisallee 19a, 04103 Leipzig, Germany
[4]Melnikov Permafrost Institute, SB RAS, 36 Merzlotnaya str., Yakutsk, Republic of Sakha, Russia, 677010
[5]BEST International Centre, North-Eastern Federal University, 58 Belinsky str., Yakutsk, Republic of Sakha, Russia, 677027

*Correspondence to:* Torben Windirsch (torben.windirsch@awi.de)

[Figure]

Figure S1 - Digital mapping of the Yukechi Alas landscape (red); lakes were digitalized in blue, Alas basins were digitalized in green; sampling locations Alas1 and YED1 marked in yellow; the 765 m distance between YED1 and Alas1 is marked by a red dotted line; satellite image taken from Google Earth.

[Figure]

Figure S2 - Age-depth model for YED1 and Alas1; radiocarbon ages in blue with uncertainties; median in red; model range indicated by grey dotted lines; created with the Bacon package in the R environment.

[Figure]

Figure S3 - C/N values of YED1 (triangles) and Alas1 (dots) plotted over $\delta^{13}C$ values.

[Figure]

Figure S4 - Sediment triangle after Shepard showing the grain size composition.

[Figure]

Figure S5 - Grain size distribution for the YED1 core; Y1 to Y4 indicate the different stratigraphical units; core unit Y2 did not hold enough sediment.

[Figure]

Figure S6 - Grain size distribution for the Alas1 core; A1 to A4 mark the different stratigraphic units.

---

## Author Response (AR1)

*The paper "Organic Carbon Characteristics in Ice-rich Permafrost in Alas and Yedoma Deposits, Central Yakutia, Siberia" by Windirsch et al. reports detailed analyses on carbon and ice contents, stable water isotopes, soil grain size distribution, and age estimates for two long ground profiles in the Central Yakutia, Russia. The presented materials are rare and highly valuable to understand landscape development and con- tents in the permafrost of the Eastern Siberia.*

> Thank you very much for acknowledging the general importance of the data presented in our paper. We are grateful for the review and acknowledge the reviewer's comments, which really improved and strengthened the paper.

*Although I expect this paper to be finally published in Biogeoscieces, the authors have not fully utilized, described, and discussed the data presented. The importance of this paper is the rareness of the sample core. I encourage the authors to enrich their descriptions about each core unit as a valuable drilling log of a permafrost region. Some portions in Discussion are not logically constructed. For the main datasets of stable water isotopes and carbon parameters, a more in-depth and quantitative discussion is anticipated.*

> Thank you very much. We worked on the point raised above. Besides detailed work on the manuscript, we added the core log as well as all core pictures to the PANGAEA database.

*Although the authors want to focus on the organic carbon characteristics as indicated in the title, more comprehensive discussion and interpretation must be done using available water geochemistry, cryostratigraphy, grain size distribution, magnetic susceptibility, etc. The usage of references or previous studies is poor and is often not clear or inappropriate.*

> As stated by the reviewer, the focus of the study has been the carbon characteristics. All other data is included to decipher the general changes of the landscapes and to have further evidence on the causes of carbon changes. We agree to add more comprehensive discussion of the background data and extended the result part and discussion for these. Major changes can be found in the result section, as well as in the discussion.

*The discussion of sedimentation rates and their changes is problematic and too speculative. Radiocarbon age of bulk soil organic matter (SOM) usually much less reliable than that of macro plant remains. In permafrost environments, old (easily more than ten thousand years)*

*carbon can co-exist in the same sedimentation stage, or younger carbon can be incorporated in a deeper layer.*

Thank you. We are aware of the dating problem and added this to the results section. Moreover we added an uncertainty statement to the section where we addressed sedimentation rates: "In addition, the dating of bulk sediments very close to the maximum datable age of ~ 50,000 yr BP may cause a high uncertainty in the absolute ages of sediment layers". The reason for taking bulk samples for dating was also extended by "We used bulk sediment samples for dating due to a lack of macro-organic remains within the deposits."

*High mobility of SOM and geomorphological processes such as thermokarst and cryoturbation give a large uncertainty to sedimentation processes. Without rigorous consideration about the validity of the age model based on bulk SOM, the discussion about the changes in sedimentation rates does not make sense.*

As stated above, we added a statement for the age model. Besides doing the statistical-based modelling, this is the best we could get out of the organic poor sample material.

*As the authors explained, the upper part of the Alas1 accumulated by Yedoma deposit thaw. The Yedoma thaw involves differential ground subsidence and mixing of ground material through the thermokarst processes, which induce destruction of original stratification. The authors should discuss the limitations of the obtained dating data and build further discussion only based on reliable information. Below, I listed individual points to be revised or to be clearly explained in the revised manuscript to achieve a publication quality.*

Thank you very much for this detailed work on improving the age discussion.

*P1-L25-26: What is "a potential theory of Holocene influence"?*

With "potential theory of Holocene influence" we tried to put emphasis on the fact that there could have been Holocene input signals in the core. In our case we saw that the water isotope signals are indicating a permanently frozen state, making this Holocene deposition unlikely.
The sentence was changed to: "Pore water isotope data from the Yedoma core indicated a continuously frozen state except for the surface sample, thereby ruling out Holocene reworking."

*P1-L30: What do you mean ". . . the Yedoma core can be duplicated."?*

Thanks for pointing to this unclear wording. This means that similar low-carbon layers were found in the Alas core as well.
We changed it to: "Similar to the Yedoma core, some sections of the Alas core were also OC poor (< 0.1 wt%) in 17 out of 28 samples."

*P1-L31-36: The authors describe using numerous "different" and "differ" works, but I suggest to revise to explain them more concretely. I could not understand how different.*

We change this as suggested:

Changed to "With its coarse sediments with low OC content (OC mean of 5.27 kg/m³), the Yedoma deposits in the Yukechi area differ from other Yedoma sites that were generally characterized by silty sediments with higher OC contents (OC mean of 19 kg/m³ for the non-ice wedge sediment)."

*L36: How the Alas core gives clear insights?*

The Alas core shows how the Yukechi Yedoma may transform under future thawing processes, as the deposits of the Alas core have once been very similar to "today's" Yedoma deposits but then started thawing. To clarify this, we have changed the sentence to: "The Alas core, strongly affected by extensive thawing processes during the Holocene, indicates a possible future pathway of ground subsidence and further OC decomposition for thawing Yakutian Yedoma deposits."

*P2-L16: Why Yedoma deposits highly vulnerable to thaw? Usually, high-ice content permafrost is more robust to thaw because of larger latent heat storage than drier permafrost. We can say Yedoma is highly susceptible to thermokarst though.*

Thank you for making this point. We clarified as follows: Yedoma is highly vulnerable to thaw induced landscape changes due to loss of excess ice. We added this to the respective sentence. You are right, high ice contents take more energy and time to thaw, but once thawed, Yedoma deposits undergo intense subsidence and reworking processes in contrast to a sediment without excess ice. And, once the ice started melting, as it is happening in current rising air temperatures, this latent heat storage will increase permafrost thawing even further by storing more heat in already melted parts, i.e. the water ponds. Also, thawing is an important part of thermokarst processes, so being vulnerable to thermokarst means being vulnerable to thaw as well.

*P2-L24: "That these processes. . ." ???*

Its clarified now to: "The occurrence of these draining and refreezing processes can usually be determined by…"

*P3-L25: active layer thickness should be larger/smaller rather than higher/lower.*

Thank you. Changed accordingly

*P4-L24: What do you mean by ". . .if 'previously' unfrozen,. . ."?*

Previously, in this case, means that these core sections were part of a talik, but were frozen for transportation, storage and subsampling. This only means, that water isotope signals from these sections must be accounted for as "pore water" and not "pore ice". We described this now more clearly as "Pore ice or, if the sediment was unfrozen during drilling, pore water was extracted using…"

*What is "artificial roots (Rhizones)"?*

We apologize for the unclear description. "Rhizones" are artificial plant roots manufactured by Rhizosphere Research Products. They are used for extracting and filtering soil water. We added further description on this to the manuscript.

*Please describe more details about the water extraction and explain how you avoid evaporation from the sample water.*

Now we described it in the following way: "In order to avoid evaporation, the samples were thawed at 4 °C inside their sample bags and sealed tightly after inserting the Rhizones."

*P4-L26: Please add information about subsample intervals.*

The subsamples were distributed over the core in 50 cm steps. We took more subsamples if there was a particularly interesting section such as abrupt sediment changes, organic remains, unusual cryostructures etc.
We specified our subsampling approach now as follows: "Subsamples were equally distributed along the cores according to visual changes in the core or at least every 50 cm in order to capture the major stratigraphic layers and its specific sediment properties."

*P4-L28: Do you have a reason to use "absolute" ice content? I think gravimetric ice content makes clearer what you are dealing with.*

We used the absolute ice content, because it is needed to calculate the bulk density. Also, it might be confusing for colleagues working outside the cryosphere to see values > 100%. To clarify this we added: "We decided for the absolute ice content as the gravimetric one, normalized with the dry sample weight, is not suitable for further calculations".

*P4-L29: I suggest to distinguish liquid and solid water more clearly. There needs more explanation to "water which froze after drilling."*

As the cores were taken in winter, freezing of liquid water in talik areas occurred after bringing these parts to the surface with air temperatures around -20 °C. Therefore, the refreezing could not be avoided. We added a description of the taliks, which are the only "liquid water" areas, in the core descriptions in chapters 4.1 and 4.2 as well as indication in the figures 3 and 5.

*P5-3.2.5: Please describe the accuracy of the stable isotope analyses.*

We added the standard deviation for all isotopic measurements.
For δ13C: "with an analytical error of ≤ 0.15 ‰."
"The analytical error for δ2H was ≤ 0.8 ‰ and for δ18O it was ≤ 0.1 ‰. The d excess was calculated as well from these values."

*P6- 3.2.6: Please explain briefly about the bootstrapping approach.*

We restructured the chapter and increased the detailed description on the bootstrapping approach: "We estimated the carbon budget of the Yukechi Alas area after Eq. (1), using a bootstrapping approach.

$$OC \ quantity \ (kt) = \frac{thickness * coverage * \frac{100 - WIV}{100} * BD * \frac{TOC}{100}}{10^3}$$

with deposit thickness in m, coverage in m², WIV in vol %, BD in kg/m³ and TOC in wt%. For all TOC values below the detection limit (0.1 wt%), a value of 0.05 wt% was set. Missing bulk density values, resulting from low ice contents (< 20 wt%) (Strauss et al., 2012), were calculated after Eq. (2), which describes the relation between TOC and bulk density in the examined cores.

$$bulk\ density = 1.3664^{-0.115 * TOC}$$

The core length of the examined cores was assumed to represent the different ground types, resulting in a deposit thickness of 22 m for Yedoma deposits and 20 m for Alas deposits. A mean wedge-ice volume of 46.3 % for Central Yakutian Yedoma deposits and 7 % for Alas deposits of Central Yakutia was assumed following Ulrich et al. (2014). We estimated the deposit coverage of Yedoma and Alas deposits using satellite imagery as shown in figure S1. The ice wedge in YED1 was excluded in the bootstrapping.

Bootstrapping calculations were done after Jongejans and Strauss (2020) for the upper 3 meters, the different core units as well as for the complete cores (Table 2) using the "boot" package in the R environment. Bootstrapping included 10,000 iterations of random sampling with replacement. We used combined BD and TOC values, as they are not independent, and corrected for irregular sampling by value replication according to depth interval. We calculated the mean and standard deviation of all iterations."

*P7-L21: What do you mean by "...dated with an infinite radiocarbon sample."?*

Sorry for the unclarity here. This means that the sample could not accurately be dated using radiocarbon dating. It was now changed to "The age of Y3 (between 1927 and 1010 cm bs) was determined as an infinite age (below 14C detection limit) of the radiocarbon sample."

*P7-L26: Please describe more about ice characteristics in this unit. Add some photos in the manuscript.*

We added a photo of the ice wedge ice to figure 3 and added a short description as follows: "Y2 (1010 to 714 cm bs) consisted of massive wedge ice, which contained very little sediment inclusions (Fig. 3b [2]). "

*P8-L26: If the active layer thickness reaches to 200 cm, the seasonal thaw depth touches the top of the talik (160cm). Does this mean the thickness of the seasonally freezing layer is 160cm?*

Yes, it does mean that the freezing layer in this specific location is 160 cm. The given 200 cm mentioned here is a general value for grassland areas in alases of Central Yakutia, following Fedorov (2006). We changed this statement as follows: "The frozen sediment of the uppermost 160 cm bs represents the seasonally freezing layer."

We added more details to this topic in the study area description: "The modern active layer thickness in Central Yakutia is approximately 1.5 m but it can be larger in grasslands, such as within Alas basins (about 2 m and more), and smaller below the taiga forest (less than 1 m) (Fedorov, 2006). For the Yukechi Alas deposits, the active layer depth can be estimated at around 200 cm and therefore reaches down into an observed talik, following Fedorov (2006). Taliks form because of a recent or already

drained lake that prevented winter freezing, or an incomplete refreezing of the active layer."

*Section 4.3: Why some YED1 points plot above the GMWL on the LEL? Could you add some explanation in Discussion? Which portion of the core are they?*

The samples you mentioned are from the lowest part of the YED1 core. Due to the permanently frozen state of the core, these isotope values show characteristics of climatic conditions from ~45,000 yr BP. We restructured this paragraph to make it more clear: "δ18O ranged between -25.16 ‰ at the lowermost sample and -30.70 ‰ at 1071.5 cm bs with a much less negative value of -15.53 ‰ closest to the surface. While the uppermost Yedoma sample had a δ²H value of -120.8 ‰, all other Yedoma samples showed much more negative values between -181.3 ‰ (2209.5 cm bs) and -221.6 ‰ (1071.5 cm bs). Values almost aligned with the global meteoric water line (GMWL) and the local evaporation line (LEL) of Central Yakutia (Wetterich et al., 2008), except for the ice wedge samples of YED1 (Fig. 7a). The isotope data obtained from the ice wedge samples had more negative values for both δ2H (-220.6 ‰ to -228.6 ‰) and δ18O (-29.58 ‰ to -30.55 ‰) in comparison to the remaining YED1 core. The d excess values are lowest in the YED1 ice wedge (lowest value of 9.3). Other values range between 3.5 in the uppermost sample as an outlier and generally range between 14.7 and 29.5 with no clear trend visible."

We also expanded the discussion on the water isotope data in section 5.2. To strengthen our water isotope expertice we invited Thomas Opel to the author team.

*P10-L27-29: Similarities between which units of them?*

This is a general statement on the similar materials found in both cores (fine, silty material and coarse, sandy material).

*P11-L2-5: Please rewrite this sentence. It is hard to understand probably because of the usage of too many conjunctions.*

We have changed this statement: "On the one hand, low TOC content could be a result of high organic matter decomposition during accumulation or during a thawed state, especially in thermokarst deposits. On the other hand, it could reflect low carbon input. The low stable carbon isotope data of our cores (between -24.06 and -27.24 ‰) are comparable to other studied sites from the Yedoma domain (Schirrmeister et al., 2013;Strauss et al., 2013;Jongejans et al., 2018) and suggest that the source signal was more or less constant with time. "

*P11-L7-8: I see varying values of d13C with depth in the profiles. What does your "constant" mean? You are comparing one data point in the active layer and data from other depth? I think it is not valid to compare decomposition rates between organic carbon currently decomposing in the active layer and ones formerly exposed to decomposition for an unknown period then frozen. As far as I see the profile, both C/N ratio and d13C fluctuate with depth even in the same unit. Is this reflected by varying decomposition rates?*

Thank you very much for this detailed comment. We changed this statement to make it more clear: "Both cores show the lowest δ13C values closest to the surface as the

organic material is the most recent and therefore least decomposed. As we look deeper in the cores, δ13C becomes higher (less negative) with no further trend over depth in the Alas and Yedoma deposits; we interpret this to mean material found at depth has already been further decomposed before freezing. Decomposition ceased when the lower samples became frozen, resulting in δ13C values not showing a clear trend at depth."

*P11-L16-20: "The similarity of the low C/N ratios from both cores. . ." Which both cores? Your YED1 and Alas1? Or are you comparing with other Yedoma sediments found near the Arctic sea? I could not understand why this similarity supports the sedimentation of organic-poor materials. Physical conditions (such as climate and hydrology) between the comparing regions are quite different.*

This is indeed a comparison between YED1 and Alas1 cores. We stated this more clearly: "From the fact that both cores, Alas and Yedoma, show low C/N ratios, …". This comparison supports the explanation of organic-poor material input for the low carbon content, as low carbon contents in syngenetic Yedoma cannot be a result of organic matter decomposition after incorporation into the permafrost. The fact, that very similar to slightly lower values are present in the Alas core, shows that both cores origin from the same material.

*P11-L22-: Please show both profiles in one graph for better comparison. The heights of 0 m of the graphs of d18O profiles differ in Fig. 7 b/c,  but not 9 m illustrated in  Fig. 2. You can reflect the relative height difference in the same graph if you want. Similarly, please add a graph showing d-excess profiles of both cores in Fig. 7 (not in supplemental figures) and discuss evaporation and freeze-out fractionation processes that might have affected to your profiles. I think Fig. S7 should be shown as the main figure, probably, combining with Fig. 7 b/c and d-excess profiles if you want to discuss dD profiles.*

Thank you. We combined figure 7 and figure S7 and added d excess plots. We rearranged all plots for better readability.

*P11-L21-26:  This entire paragraph is not logically described and I do not find sounding discussion. Why the age inversion and stable water isotope signals are related? Why particularly 35 K BP is the timing of the shift in sediment input? "This would have been . . ." What "This" indicates here?*

We restructured this paragraph and made it clearer. We also added further discussion in the paragraph that now reads as follows: "We found age inversions in both cores with similar age and depth (YED1 49,232 cal yr BP, 1998.5 cm bs; Alas1 42,865 cal yr BP, 1967.5 cm bs) (Fig. 4 and 6, Fig. S2). While cryoturbation might seem an obvious explanation, we suggest that this process did not play a major role here due to the long-term frozen state of YED1. Rather, we assume that the age inversions indicate a temporary shift in sediment input at approximately 35,000 cal yr BP causing some deposit reworking in the watershed and the incorporation of older material into younger sediments. In addition, the dating of bulk sediments very close to the maximum datable age of ~ 50,000 yr BP may cause a high uncertainty in the absolute ages of sediment layers (Reimer et al., 2013). Therefore, the rather small age inversions (> 49.000 cal yr BP to 49,232 cal yr BP in YED1, and 45,870 cal yr BP to 42,865 cal yr BP in Alas1)

could be a result of material mixture in dated bulk samples. The radiocarbon ages above this age inversion align well with a simulated sedimentation rate, as shown in figure S2."

*P11-L27: "permanently" means always and forever. You should assign a more specified period of time when you discuss topics like in your manuscript. It is impossible to be in a permanently frozen state throughout the depositional history.*

Thank you. We changed this to "perennially frozen conditions since incorporation into permafrost". Of course, after deposition this material has been part of the active layer for a period of time. We hope this is more specific now.

*P11-L28: Please clarify depths for "the lower Yedoma pore ice" and "the uppermost sample." "the uppermost sample" is the data at 0m in YED1?*

We added the sample depth to the description, and as only the uppermost signal is very different from the remaining core, we removed "lower" from the description.
"...much lower $\delta 18O$ values for the Yedoma pore ice in comparison to the uppermost sample (4 cm bs in YED1) showing a water isotope signal…"

*P11-L31: "The very similar.." to what?*

We removed this sentence.

*P12-L1: The original data were from Popp et al. (2006)?*

Changed accordingly

*P12-L2: "offset" of what? Your ice-wedge values against what?*

We added detailed explanation to this paragraph as follows: "The stable isotope ratio values of ice wedges (mean $\delta 18O$ of -30 ‰, mean $\delta 2H$ of -224 ‰) reflect winter precipitation and fit well into the regional pattern for MIS 3 ice wedges in Central and Interior Yakutia (Popp et al., 2006;Opel et at., 2019) while the d excess shows a much elevated value (16 ‰) compared to the regional pattern (Popp et al., 2006;Opel et al., 2019). It should be noted that the d excess values from the central core parts correspond well to the regional values from Mamontova Gora, Tanda and Batagay (Opel et at., 2019), while the others fit to those of the host sediments and are potentially overprinted by exchange processes between wedge ice and pore ice (Meyer et al., 2010). Due to the low number of datapoints, no meaningful co-isotopic regression could be calculated. The stable isotope composition of pore ice shows a co-isotopic regression of $\delta 2H = 6.61 \, \delta 18O - 18.0$ ($R^2 = 0.97$, n = 23), which is typical for Yedoma intrasedimental ice (Wetterich et al., 2011;2014;2016). The isotope values plot well above the GMWL of the cold season (Papina et al., 2017) suggesting a substantial proportion of (early) winter precipitation (usually characterized by high d excess values) for the pore ice, which is also evident for some units of the Batagay megaslump (Opel et al., 2019). The decreasing upward trend of pore ice isotopic $\delta$ values might point to a general cooling in Central Yakutia during the covered time span of our study. However, as it is accompanied by an opposite increasing trend in d excess, these

values may be overprinted by secondary freeze-thaw processes in the active layer and rather reflect the strength of these fractionation processes (Wetterich et al., 2014)."

*P12-L4: As I mentioned above, the age constraints around 40-50 K BP need reconsideration. Furthermore, isotope signals of pore water do not simply show climate change. Lots of complicated fractionations and changes in seasonal meteoric water components could affect the signals.*

> Please see our comment before.

*P12-L5-: This topic sentence doesn't make sense to me from your discussion above. What do you mean "lake converage"? Your discussion mixes decomposition rates during Yedoma deposition and during alas formation.*

> We restructured and clarified the whole paragraph.

*P12-L25-: Cryostructure of frozen sediments mainly depends on soil grain size distribution, freezing rate, and moisture content. Sandy sediments usually have fewer pore spaces comparing to sediments consist of finer soil particles. Sandy sediments tend to have structureless cryostructure because sandy particles have a limited amount of unfrozen water under cryotic conditions.*

> We appreciate this comment and further explanation. Also, we do not argue with this explanation, as this does not oppose our explanation of refreezing resulting in structureless pore ice due to the sandy sediment.

*Section 5.2 the first paragraph: I think the authors are misunderstanding or misinterpreting the papers Wetterich et al. (2009), French and Shur (2010), and Iwahana et al. (2014) as to the mechanism of cryostructure formation and sediment/carbon movements in the ground.*

> We do not fully understand the reviewer's comment here. The discussions in this paragraph primarily relate to the visually visible cryostructures and the conclusions that can be drawn from them about the freezing processes after sedimentation. Based on our data, we see clear similarities in the cryostructures to those described by Wetterich et al. and French and Shur. In order to avoid further misunderstandings, we have rewritten the paragraph slightly and adjusted the references.

*Section 5.2 the second paragraph: Discussion in this paragraph makes sense to me. This discussion should be placed in Section 5.1 as it is tightly related to carbon accumulation and loss at this site.*

> We moved this part to section 5.1 as suggested

*P13-L23: Why "organic-bearing"? I thought you are discussing the transportation of organic-poor sediments. Could you discuss this sandy deposit could be Aeolian or not using your grain size data?*

> Thank you for your comment. We decided to delete this sentence as is not needed here to support the deposition of organic-poor sediments.

*P13-L25: Where the "7000 yr" came from? Aeolian materials cannot be organic-poor?*

> The 7000 yr originate from the age difference of the radiocarbon ages below and above this sandy Y3 unit. We tried to clarify this as follows: "From our data we see that the sandy layers were deposited in approximately 7000 yr (radiocarbon dates below and above these layers). As the sediments are rather coarse (115.3 µm mean grain size), a fluvial deposition is more likely than an aeolian deposition (Strauss et al., 2012)."
> Of course, aeolian materials can be organic-poor, too, but this material is rather coarse at the same time which, in our opinion, makes fluvial transport a more suitable explanation.

*P13-L29: What do you want to refer to Diekmann et al. (2017)? The climate changes?*

> Yes, as Diekmann et al. sum up the available literature on Siberian climate very well. We added more literature to this topic. To improve clarity we rearranged this sentence to "The climatic changes (Diekmann et al., 2017;Murton et al., 2017) and the resulting…"

*P14-L4: What the "this" indicate? The deposition of silty organic-bearing material on top of the sandy layers, decreased water availability, or a climatic backshift to colder conditions? In any case, I could not understand the logic.*

> We apologize for this mistake, the sentences lost their correct order during internal revision. We now rearranged the sentences to: "The climatic changes (Diekmann et al., 2017; Murton et al., 2017) and the resulting higher water availability during the deposition period of the sandy layers may have caused changes in fluvial patterns on the Abalakh Terrace. More water could cause higher flow velocities under warmer climatic conditions and therefore increased erosive power, leading to the formation of new flow channels (Reineck and Singh, 1980). With a climatic backshift to colder conditions during MIS 2, water availability decreased and silty organic-bearing material was deposited by seasonal flooding again on top of the sandy layers. This likely led to lake initiation on top of the Yedoma deposits. The underlying ground began to thaw and subside, forming the Yukechi Alas basin. During this process, ice was lost from the sediment and the ground subsided by at least 9 m (height difference of 9 m between YED1 and Alas1 surface). Surface or lake water was able to percolate through the unfrozen sediments. This becomes visible by the homogeneous water isotope signal similar to the YED1 surface sample (Fig. 7). Under the unfrozen aquatic conditions in the sediment, microbial activity started, resulting in the decomposition of the already small amount of organic material (Cole et al., 2001). When the lake drained, the sediments started to refreeze both upward from the underlying permafrost and downward from the surface, leaving a talik in between (Fig. 5)."

*P14-L5: Why you refer to Grosse et al. (2013) for "The underlying ground began to thaw"??*

> We decided to remove this citation here.

*P14-L7: "as visible in the . . ." should be explained concretely.*

We added further explanation by: "Surface or lake water was able to percolate through the unfrozen sediments. This becomes visible by the homogeneous water isotope signal similar to the YED1 surface sample (Fig. 7)."

*P14-L12: "shrunk . . . from . . . 2200 to 1200cm..." Please discuss the possibility of subsidence using excess ice volume information. Is there enough excess ice content in the initial Yedoma to reduce its volume from 2200 to 1200cm thick?*

As we have no field data on the distribution of ice wedges in the Yukechi area, we cannot exactly say if the ice volume would be sufficient for this decrease. Though, the matching characteristics of the stratigraphic units of YED1 and Alas1 and the existence of large ice wedges in this area as reported already by Russian scientists before support the hypothesis of the stated ground subsidence. To clarify this, we added the following text: "The presence of large ice wedges in the area supports this theory of ground subsidence during thaw as it hints on large excess ice contents of the ground (see Fig. S7) (Soloviev, 1959)."

*P14-L12-13: Again, what "this" indicates here?*

Changed to "These subsidence processes" to be more specific about relations in this paragraph.

*P14-L15-16: I don't understand why you started to discuss the tipping point suddenly. Is this relevant and supported by your results?*

We decided to remove this sentence from the discussion.

*Section 5.3 the first paragraph: How the 4.40 kg/m3 was derived? Please explain the relation to 4.48 and 6.93 for the two sites?*

This number was calculated by upscaling the bootstrapping results to the study area. We added some more explanation here as follows: "Using a bootstrapping approach (Jongejans and Strauss, 2020) we found a much lower organic carbon density of 4.48 ± 1.43 kg/m3 for the top 3 m of the YED1 core. For Alas1, an organic carbon density of 6.93 ± 2.90 kg/m3 was calculated for the top 3 m. Taking the area covered by each deposit type within the Yukechi Alas landscape into account and the ice wedge volumes estimated by Ulrich et al. (2014) (see Section 2 and Fig. S2), this results in a mean organic carbon density of only 4.40 kg/m3 for the top 3 m of dry soil at the Yukechi study site. This landscape scale carbon stock density includes the entire study area (1.4 km²), including all water bodies (approximately 0.18 km²), which we assumed to contain no soil carbon"

*This could be the first carbon stock measurement below 3m for two sites in the Central Yakutian Yedoma in English literature, but have you checked Russian literature?*

We checked on international available and peer-reviewed Russian literature and added some more literature to our manuscript. We avoid citing non-peer-reviewed literature and literature that is not available to an international scientific audience though.

*Please discuss how representative your two cores for the Central Yakutian Yedoma.*

> Thank you, we expanded the discussion accordingly an concluded that our cores seem not representative for other examined Yedoma deposits from further north. However, there are no data available on Central Yakutian deep Yedoma cores in available peer-reviewed literature, which would be necessary for comparison inside this southernmost Yedoma domain. A corresponding paragraph in the conclusions chapter reads as follows: "Hence, the studied Yukechi Yedoma deposits store less carbon than other, comparable Yedoma Ice Complex deposits in the Central Yakutian area. However, there have been comparatively few studies on this so far. The biogeochemical impact of permafrost thawing in the Yukechi area might therefore be smaller than generally assumed for Yedoma deposits, as this area does not feature the high carbon stock estimates and high ice contents of other previously studied localities in Central Yakutia and elsewhere in the Arctic."

*P16-L1-2: I could not understand what you wanted to indicate from this sentence. What do you mean by "drainage"? How did the alas core reveal its composition and stratigraphy before lake formation?*

> We apologize for the misunderstanding and the ambiguities here. The grain size and carbon characteristics of the Alas core can be explained by thawing and subsiding of the local Yedoma, starting from the characteristics found in our YED1 core. Therefore, we can assume that, previous to all thermokarst processes, the deposits found in Alas1 were basically the same as the deposits in the YED1 core are today. We changed the text for more clarity to the following: "The permafrost characteristics found in the Alas core reveal that its composition and stratigraphy before lake formation and disappearance was very similar to the Yedoma core material. Its past development including thaw, the loss of old ice and surface subsidence, along with sediment compaction, shows a possible pathway for the Central Yakutian Yedoma deposits under the influence of global climate change."

*Fig. 2: Yedoma lake should be thermokarst lake? Or by local name, "dyuyodya" (Soloviev, 1973) or "dyede" (Crate et al., 2017)? Soloviev, P. A. (1973), Thermokarst phenomena and landforms due to frost heaving in Central Yakutia, Biuletyn Peryglacjalny 23, 135-155. Crate, S., et al. (2017), Permafrost livelihoods: A transdisciplinary review and analy- sis of thermokarst-based systems of indigenous land use, Anthropocene, 18, 89-104, doi: https://doi.org/10.1016/k.ancene.2017.06.001.*

> Indeed, both lakes are actually thermokarst lakes in different evolutionary stages. In order to distinguish between the two lakes found at our study site, they were named after the deposit type they were found on (Yedoma and Alas). This was also based on Ulrich et al. 2017. Differences in behaviour and distribution of permafrost-related lakes in Central Yakutia and their response to climatic drivers, Water Resources Research, 1167-1188, https://doi.org/10.1016/k.ancene.2017.06.001.
> However, we add a note in the figure caption as follows: "the terms "Alas lake" and "Yedoma lake" are chosen after Ulrich et al., 2017a in accordance to the deposit type in which the thermokarst lakes are located; following Crate et al., (2017), the Yedoma lake can also be called "dyede" due to its development stage."

*Fig. 3: Include photos of each unit including ice-wedge layer. Put "YED1" in the figure.*

> changed accordingly

*Fig. 5: Please use the same white balance and adjust the exposure/brightness of the core photos as in Fig. 3. Include core photos of the rest two units too. Put "Alas1" in the figure.*

> changed accordingly; We adjusted the brightness as far as possible without corrupting the image quality and visual appearance of the core.

*Fig. 4 & 6: Please used different markers for absolute ice content and bulk density so that people without color printer environment or color-blind can easily distinguish the two. Large caption YED1 and Alas1 within the figures would be helpful.*

> changed accordingly

*Fig. 7 & S7: Combine Fig 7 b/c and S7 and add a graph of d excess. Please use open triangle markers for ice-wedge points. The half-open triangle hinders instant discrimination between markers. Add labels "YED1" or "Alas1" on top of each graph.*

> changed accordingly; We combined figure 7 and figure S7 and added d-excess plots. For better readability, we rearranged all plots.

*Fig. S2: The blue markers with age uncertainties are not discernible. Please revise them for a clearer presentation. I do not think the information on the model confidence by black shades is meaningful and necessary in your radiocarbon age of soil organic matter. For all graphs with depth axis: Please use the meter scale, not cm.*

> graphics changed accordingly; We will continue to use cm as we go down to 1 cm step accuracy, which is much easier to read in cm units than with a meter scale. We are aware that meter is the SI unit.

*Interactive comment on*

**"Organic Carbon Characteristics in Ice-rich Permafrost in Alas and Yedoma Deposits, Central Yakutia, Siberia"**

*by* **Torben Windirsch et al.**

**Anonymous Referee #2**

*The manuscript by Windirsch et al. "Organic Carbon Characteristics in Ice-rich Permafrost in Alas and Yedoma Deposits, Central Yakutia, Siberia" examined and compared the organic carbon storage and characteristics from two rare and deep cores using different methods (C%, soil texture, 14C age, ice content, etc..). Since such deep cores are very rare, this study is very important and gives us valuable inside information about the history of these deep deposits.*

Thank you very much for this general statement of importance of our study.

*The scientific question and the used methods are well established, however, there is still room for some improvement. Overall, the discussion section is rather weak and speculative, and the based conclusions too shaky. One reason, most of the used methods were nor fully incorporated in the discussion section. Some results as for example 14C inversions or magnetic susceptibility are not really discussed and explained.*

We are grateful for the review and acknowledge the reviewer's comments, which really improve and strengthened the paper.
As suggested by the reviewer we extended and clarified the discussion and strengthened the conclusion.
Please see our comments below.

*Nitrogen data not presented, even though obviously available.*

They have not been shown as most values did not reach detection limit. This has now been referenced and implemented in the result sections 4.1 and 4.2. Moreover, the data presented in the manuscript are available on the PANGAEA repository.

*Conclusion to short, missing main points as for example that the global estimate of SOC in Yedoma might be by far overestimated and possibly not so vulnerable due to rather low ice contents*

Yedoma ice contents are low in this specific case, which is described in the conclusions. With one site we can not revise circumarctic estimates. Therefore, general Yedoma vulnerability is not the topic of this discussion. As the drained thermokarst basins are 9 m deeper there is a substantial potential for Yedoma for severe changes. Using this study to revise studies basing on a Yedoma wide study site set (Strauss et al 2017, Schirrmeister et al 2011) is not appropriate to our mind.

But nevertheless, as we state we make this point in the discussion but this is not one of the main conclusions.

*Also, recent publications suggest Yedoma being not extremely vulnerable. Missing older Russian Literature*

We checked on international available and peer-reviewed Russian literature and added some more suitable literature. We avoid citing non-peer-reviewed literature and literature that is not available to an international scientific audience. We would be happy to include more literature. Please use the comment function to add specific literature.

Besides this, we concluded that Yedoma is highly vulnerable to thaw induced landscape changes due to loss of excess ice. However, as stated by Kuhry et al, the carbon stored in Yedoma deposits might not be highly vulnerable.

We now implemented Kuhry et al. (2020) to the paper which was not published at the first submission of our paper.

*Below I list more specific points, which should be addressed before final publication.*

*P.1 L. 15:"has not yet", actually parts of Yedoma are already in the active carbon cycle;*

We changed this to "this carbon becomes available to the recent carbon cycle.", as some of it is still frozen and immobilized.

*P1 L: 30"very carbon poor", please add number;*

Thank you, changed accordingly: We added "(< 0.1 wt%)" to the description, which is the detection limit for measuring the total organic carbon content.

*P2 L2 "provide"???, and why is it important;*

We changed this to "Permafrost deposits present one of the largest terrestrial carbon reservoirs."

*P2 L6, reference on co2 in the atmosphere not appropriate. Use actual data source;*

We changed the reference to 868 Gt which was calculated using the CO2 concentration of 407 ppm, measured in 2018. This was calculated using the conversion formula given by Friedlingstein et al (2019). Global carbon budget using a factor of 2.124 for conversion between ppm CO2 and Gt carbon after Ballantyne et al (2012) for well-mixed atmospheric conditions. The text now reads: "The estimated amount of frozen and unfrozen carbon stored in the terrestrial permafrost region is 1330 to 1580 gigatons (Gt) (Hugelius et al., 2014;Schuur et al., 2015), which is up to 45% more than what is currently present in the atmosphere (~ 864 Gt, based on 407 ppm CO2 measured in 2018) (Ballantyne et al., 2012; Friedlingstein et al., 2019)."

*P2 L16 "vulnerable" recent paper by Kuhry et al. states that Yedoma is rather stable.;*

Concerning landscape vulnerability: the general high ice content in Yedoma deposits makes them vulnerable to dramatic landscape changes caused by thaw under climate warming;

Concerning C vulnerability: Kuhry et al (2020) does not state the opposite but concludes that thaw vulnerability does not mean a high vulnerability/lability of SOM. This sentence was changed as follows: "In the context of global climate change, such high ice content with pore ice and syngenetic ice wedges render Yedoma deposits highly susceptible to thaw induced landscape changes (Schirrmeister et al., 2013) and ground volume loss causing surface subsidence."

*P2 L20 Is this the case only for carbon stored within Yedoma deposits???;*

Thank you, we changed "Yedoma deposits" to "permafrost".

*P2 L21 released how? I guess you mean as carbon dioxide.;*

We specified this by adding "... in form of gases such as carbon dioxide or methane,..."

*P2 L25, please add ref.;*

We added "(Strauss et al., 2013)" to the corresponding sentence.

*P2 L27 Explain why below 3m less understood?*

This is due to the fact that only very few studies have been examining long permafrost cores. We clarified that in the text: "...especially below 3 m, are still poorly studied, as only few studies examining long Siberian permafrost cores have been conducted (Zimov et al., 2006;Strauss et al., 2013;Shmelev et al., 2017)."

*P2 L29 missing space;*

changed as suggested

*P2 L32 "822 Gt" this is wrong, the number refers not only to Yedoma, but refers to combined permafrost SOC in soils (0–3 m), deltaic alluvium and Yedoma region sediments.*

Thank you for notifying. We changed it to „permafrost region"

*P3 L5 Your 'only' 2 research questions are going under in the paragraph. Please make them stand out.*

We made the research questions more visible by adjusting the paragraphs.

*P3 L24 "in Siberia", is there no data for MAAT at least for Central Yakutia?*

Thank you for your suggestion. We changed this to "The contemporary mean annual air temperature in Central Yakutia (measured at Yakutsk Meteorological Station) is -9.7 °C."

*P3 L27, No more data or drilling campaigns since Soloviev, 1973?*

There are actually few accessible studies on the thickness of the Yedoma in Central Yakutia. The most cited studies are indeed Soloviev, 1959 and 1973.
We changed this statement and include both references as follows: "The Yedoma deposits in this region can be more than 30 m in thickness as was already shown by older Russian works (Soloviev, 1959, 1973)."

*P4 L23 "approximately every 50cm? Why this distance? Why not using visual changes in the core for the increments?*

The cores were indeed sub-sampled after visible stratigraphic changes. However, larger homogeneous sections were sampled approximately every 50 cm. We specified our subsampling approach as follows "Subsamples were equally distributed along the cores. According to visual changes, we covered all visible stratigraphic layers and we sampled at least every 50 cm in order to capture specific sediment properties."

*P6 L4 check paragraph spacing.*

Thank you, we adjusted it.

*P6 Since the main question is carbon, a chapter on how carbon estimates were calculated would be more then appropriate.*

Thank you. The methods of our carbon budget calculation are explained in chapter 3.2.6. We renamed the chapter title to "Statistics and bootstrapping approach for carbon budget estimations" to make this clearer.

*P7L18 "material composition???" soil texture.*

Changed accordingly

*P9 L6 please remove rather, enough with the given SD.*

Changed accordingly

*P9, when you compare the grain sizes, you change between median and mean, why?*

We corrected this and now use the mean for all data.

*P10 L1 I don't fully understand the SOC calculations. How many samples were used for boot strapping? Please add a table comparing these two cores with SOC data, TN data, DBD data, to 3 m meter, to the different used units and to the total.*

Thank you. We expanded the explanation on the bootstrapping approach. Also, we added a table to show SOC contents for the individual core units, based on the bootstrapping results. As TN data are not used in the bootstrapping approach, we did not include them in table 2. These data are shown in the PANGAEA repository as most values are below the detection limit. Please see our comment further above.

*P11L1-5. And what exactly are you arguing? Also, a lot of speculations. Please stick to the point. Also, the rather obvious reason for the low C comes first on page 13!*

> This was a misuse of "arguing", we apologize. We changed the sentence into: "On the one hand, low TOC content could be a result of high organic matter decomposition during accumulation or during a thawed state, especially in thermokarst deposits. On the other hand, this could reflect low carbon inputs. The low stable carbon isotope data of our cores (between -24.06 and -27.24 ‰) are comparable to other studied sites from the Yedoma domain (Schirrmeister et al., 2013;Strauss et al., 2013;Jongejans et al., 2018), and suggest that the source signal was more or less constant with time.". The reason for low C is discussed in more detail later in chapter 5.2.

*P11L13-15 But then the frozen sections should have the same C/N ratio, but they don't?*

> They do not have the same C/N ratio, as the Alas1 core was once thawed completely, as far as we conclude from the water isotope data. To highlight this we change this part as follows: "In comparison to the mean C/N ratio of 10 in YED1, the mean C/N ratio of 8 for Alas1 may indicate that the Alas deposits were only slightly more affected by decomposition due to a temporary thawed state. The development of a talik in Alas1 probably led to increased decomposition of these sediment sections and thus leading to a reduced C/N ratio. As the carbon was freeze-locked in the YED1 core and therefore was not decomposed since deposition, this similarity in mean C/N ratios indicates that the original carbon state for the deposits at both coring sites were similar, assuming the same carbon source for both deposits."

*P11 L18 "decomposed" if, then rather pre-decomposed.*

> Changed accordingly

*P11 L21-26 Ok, so what is the reason for inversions? Section by far too short. And no discussion on the other dates.*

> We restructured this paragraph and clarified. We also added further discussion in the paragraph that now reads as follows: "We found age inversions in both cores with similar age and depth (YED1 49,232 cal yr BP, 1998.5 cm bs; Alas1 42,865 cal yr BP, 1967.5 cm bs) (Fig. 4 and 6, Fig. S2). While cryoturbation might seem an obvious explanation, we suggest that this process did not play a major role here due to the long-term frozen state of YED1. Rather, we assume that the age inversions indicate a temporary shift in sediment input at approximately 35,000 cal yr BP causing some deposit reworking in the watershed and the incorporation of older material into younger sediments. In addition, the dating of bulk sediments very close to the maximum datable age of ~ 50,000 yr BP may cause a high uncertainty in the absolute ages of sediment layers (Reimer et al., 2013). Therefore, the rather small age inversions (> 49.000 cal yr BP to 49,232 cal yr BP in YED1, and 45,870 cal yr BP to 42,865 cal yr BP in Alas1) could be a result of material mixture in dated bulk samples. The radiocarbon ages above this age inversion align well with a simulated sedimentation rate, as shown in figure S2."

*P11 L32 "indicate colder climate conditions" How, and what does it mean for the core? Any refs for that?*

We deleted this sentence, as it was unnecessary information without important implications for the core.

*P12 L2 "explained by climatic variations" Is this an assumption or is there data? Need refs for this statement.*

We restructured the whole paragraph and added more detail on isotope data and d excess: "The stable isotope ratio values of ice wedges (mean δ18O of -30 ‰, mean δ2H of -224 ‰) reflect winter precipitation and fit well into the regional pattern for MIS 3 ice wedges in Central and Interior Yakutia (Popp et al., 2006;Opel et at., 2019) while the d excess shows a much elevated value (16 ‰) compared to the regional pattern (Popp et al., 2006;Opel et al., 2019). It should be noted that the d excess values from the central core parts correspond well to the regional values from Mamontova Gora, Tanda and Batagay (Opel et at., 2019), while the others fit to those of the host sediments and are potentially overprinted by exchange processes between wedge ice and pore ice (Meyer et al., 2010). Due to the low number of datapoints, no meaningful co-isotopic regression could be calculated. The stable isotope composition of pore ice shows a co-isotopic regression of δ2H = 6.61 δ18O - 18.0 (R² = 0.97, n = 23), which is typical for Yedoma intrasedimental ice (Wetterich et al., 2011;2014;2016). The isotope values plot well above the GMWL of the cold season (Papina et al., 2017) suggesting a substantial proportion of (early) winter precipitation (usually characterized by high d excess values) for the pore ice, which is also evident for some units of the Batagay megaslump (Opel et al., 2019). The decreasing upward trend of pore ice isotopic δ values might point to a general cooling in Central Yakutia during the covered time span of our study. However, as it is accompanied by an opposite increasing trend in d excess, these values may be overprinted by secondary freeze-thaw processes in the active layer and rather reflect the strength of these fractionation processes (Wetterich et al., 2014)." Moreover, we strengthened our stable water isotope expertise by inviting Thomas Opel to the author team.

*P13 The final reason for the low C% should be mentioned earlier. Also, this page can be shortened.*

Thank you for this comment. We agree and moved this interpretation into section 5.1. We also restructured and shortened this chapter.

*P15 L15 & 23, repetitive sentence. Also, main conclusion, Yedoma SOC estimates likely too high.*

We removed the repetition. Thanks for this comment but following Schneider von Deimling et al (2015) the SOC estimations for Yedoma could still be correct, giving the Yukechi site as a low-carbon example. Also, due to the sediment diversity within the YED1 and Alas1 cores, the Yukechi Alas landscape does not represent a typical Yedoma composition.

*P15 L28. "high ice content.... vulnerable" actually the opposite. You showed very little ice content except one one ice wedge. Yedoma "had an estimated" ice wedge and lenses content of up to 90%. These two cores have far less. So these Yedoma deposits are not very vulnerable.*

The high volumetric ice content in Yedoma deposits is mainly due to the presence of large ice wedges in the area (please see new figure S7), as in this case in our Yedoma core. We have chosen the Yedoma coring location in the centre of a polygon for not hitting an ice wedge. In this context, the Alas core was not included, as it represents already reworked Yedoma deposits in which the Pleistocene ice has largely melted. The pore ice content of the Central Yakutian Yedoma sediment deposits is comparatively smaller, but the sensitivity of the sediments to thawing processes is retained as for the ice wedges. This is also shown by the numerous large thermokarst landscape features in the region, but also by the currently observed very rapid thawing processes.

[revised manuscript text omitted]

---

## Referee Report (RR1)

The manuscript by Windirsch et al. "Organic Carbon Characteristics in Ice-rich Permafrost in Alas and Yedoma Deposits, Central Yakutia, Siberia" reports detailed analyses on organic carbon storage, grain size and other characteristics from two different deep cores. Such data is rare and valuable as it helps to understand the landscape development from that area. Especially, since it points out the heterogeneity within Yedoma deposits.

Overall, I am very happy with the revised version of the manuscript. However, the discussion section can still be improved. Partly, still not logically constructed with repeated information. Also, the language in the discussion section needs improvement (wrong usage of tenses etc. makes the reading complicated.
Abstract is too long and, in the beginning, more of an introduction.

Below only a few minor comments.

Page 2, Line 2-4: In comparison to the total amount of C stored in permafrost is Yedoma's amount not particular large. I would suggest to remove 'In particular'.
Page 2, Line 3-4: 'After thaw, this carbon becomes available to the recent carbon cycle'. becomes part of the ...,  Or just remove this sentence
Page 12, Line 19: 'We interpret….' Please rewrite this sentence. Also, there is no higher decomposed, more decomposed or higher decomposability
Page 12, Line 20: 'when the deposits became frozen…', once the deposits froze
Page 13, Line 34. Please use Deuterium instead the d, also in other places.

Figure 1a, please add a scale bar to the figure.
Figure S2, please add (a) and (b) for the two different sites

---

## Author Response (AR2)

**Author's Responses to Minor Revisions on**

**"Organic Carbon Characteristics in Ice-rich Permafrost in Alas and Yedoma Deposits, Central Yakutia, Siberia" *by* Torben Windirsch et al.**

**Referee #1 - Go Iwahana**

Thank you for your thorough responses to my comments. The manuscript is now very informative and became suitable for publication. I believe this study provided highly valuable findings with datasets and discussion on the evolution and carbon storage of Yedoma in the Central Yakutia, hence important insights on the overall understanding of permafrost dynamics. Finally, I would like to suggest a minor addition to the explanation of the bootstrapping approach as below.

Thank you very much for this encouraging evaluation of our work!

P6- 3.2.6: Please explain briefly about the bootstrapping approach. (P.9 L.7) Thank you for adding the explanation of the bootstrapping approach. However, I have several questions here.

My suggestion was adding a sentence or two to explain a concept of the method, so readers need not go to the reference. We appreciate a brief summary of the concept of the method. The added explanation is not so kind to readers (for example, why "10,000 iterations of random sampling (of what?) with replacement" was conducted?).

We added some more in-depth explanation on the statistical method itself: "Bootstrapping is a statistical method to estimate the sample distribution using resampling and replacement (Crawley, 2015). Resampling consists of drawing randomly selected samples from the dataset (i.e. BD and TOC) repeatedly (10,000 iterations), after which those values are fed into the formula. Replacement refers to the fact that the drawn samples in each iteration are available for all following iterations. We used combined BD and TOC values, as they are not independent. In addition, we corrected for irregular sampling by value replication according to depth interval so that values spanning larger intervals have a higher chance of being drawn. We calculated the mean and standard deviation of all iterations."

**You added the explain with the equation, but I do not still understand what is WIV. I guess it is Wedge Ice Volume?, but I could not find a description of how it was determined.**

We changed the description of formula 1 to now read "wedge-ice volume (WIV)". To help readers, we added information to the wedge-ice volume determination: " A mean wedge-ice volume of 46.3 % for Central Yakutian Yedoma deposits and 7 % for Alas deposits of Central Yakutia was assumed following Ulrich et al. (2014) who determined average wedge-ice volumes for several deposit types in multiple locations in Siberia."

Why "missing bulk density values" was resulted from low ice contents? According to Strauss et al. (2012), it seems to be related to ice saturation. Anyway, it is not clear what "missing bulk density values indicate" and from which depth (and which core) are those values here.

Following Strauss et al. (2012) it is indeed necessary to have fully ice-saturated sediment to be able to calculate bulk densities, as otherwise empty pore space has to be assumed, distorting the results. The key number for this calculation is an absolute ice content of at least 20 wt% to ensure full saturation. As we were not able to directly measure bulk densities, we had to calculate them following Strauss et al. (2012) which therefore was not possible for samples with absolute ice content < 20 wt%. We added some more details to the description and referred to the measurement data available via PANGAEA: "Missing bulk density values, resulting from low ice contents (< 20 wt%) and therefore not fully ice-saturated sediments (Strauss et al., 2012), were calculated after Eq. (2), which describes the relation between TOC and bulk density in the examined cores. This had to be done for 9 samples in YED1 and 12 samples in Alas1 (see also Windirsch et al. (2019))." We decided not to name all affected samples here due to the wide spread across the cores and only the minority of bulk density values being calculated in each core.

**Anonymous Referee #2**

Overall, I am very happy with the revised version of the manuscript. However, the discussion section can still be improved. Partly, still not logically constructed with repeated information. Also, the language in the discussion section needs improvement (wrong usage of tenses etc. makes the reading complicated.

Thank you very much for this comment and the appreciation of our efforts. We revised the discussion again and tried to make it easier to follow removing some unnecessary repetitions. We also tried to unify and correct the tenses used.

Abstract is too long and, in the beginning, more of an introduction.

We revised and shortened the abstract.

Page 2, Line 2-4: In comparison to the total amount of C stored in permafrost is Yedoma's amount not particular large. I would suggest to remove 'In particular'.

Thank you! We changed this accordinly.

Page 2, Line 3-4: 'After thaw, this carbon becomes available to the recent carbon cycle'. becomes part of the ..., Or just remove this sentence

We decided to remove this sentence in our effort to shorten the abstract.

Page 12, Line 19: 'We interpret....' Please rewrite this sentence. Also, there is no higher decomposed, more decomposed or higher decomposability

We changed this sentence to "This indicates that the present material was already further decomposed when becoming frozen."

Page 12, Line 20: 'when the deposits became frozen...', once the deposits froze Thank you; changed accordingly

Page 13, Line 34. Please use Deuterium instead the d, also in other places.

We decided to stick with "d excess", as this term is widely used in literature. However, we introduced "d excess" as an abreviation of "Deuterium excess" in the first use.

Figure 1a, please add a scale bar to the figure. We added a scalebar to the figure.

Figure S2, please add (a) and (b) for the two different sites changed accordingly

**Organic Carbon Characteristics in Ice-rich Permafrost in Alas and Yedoma Deposits, Central Yakutia, Siberia**

Torben Windirsch1,2, Guido Grosse1,2, Mathias Ulrich3, Lutz Schirrmeister1, Alexander N. Fedorov4,5, Pavel Ya. Konstantinov4, Matthias Fuchs1, Loeka L. Jongejans1,2, Juliane Wolter1, Thomas Opel1 and Jens Strauss1

[revised manuscript text omitted]

\* calibrated using Calib 7.1 (Stuiver et al., 2018) equipped with IntCal 13 (Reimer et al., 2013)

Table 2 – SOC contents for the individual core units, based on the bootstrapping results; calculations were done for 1  $m^2$ ; the measurement data used in the bootstrapping approach (bulk density, TOC density) are provided in the data sheet in the PANGAEA repository; \* refers to samples with TOC content < 0.1 wt%; for organic carbon pool calculations, we assumed a TOC of 0.05 wt% for these samples; note: we excluded unit Y2 in the calculations.

| Core  | Depth [cm bs]         | Number of samples used in bootstrapping | Mean dry bulk density
[10³ kg/m³] | Mean TOC content [wt%] | Mean SOC content
(bootstrapping results)
[kg/m 3 ] |
|-------|-----------------------|-----------------------------------------|--------------------------------------|------------------------|---------------------------------------------------------------------|
| YED1  | 0 – 300               | 7                                       | 1190                                 | 0.42                   | $4.48 \pm 1.43$                                                     |
|       | 0 – 714 (unit Y1)     | 13                                      | 1090                                 | 0.59                   | $8.31 \pm 1.41$                                                     |
|       | 1010 – 1927 (unit Y3) | 18                                      | 1172                                 | 0.10                   | $0.86\pm0.32$                                                       |
|       | 1927 – 2235 (unit Y4) | 5                                       | 910                                  | 1.14                   | $11.50 \pm 1.36$                                                    |
|       | total core            | 36                                      | 1105                                 | 0.46                   | $5.27 \pm 1.42$                                                     |
| Alas1 | 0 - 300               | 5                                       | 1257                                 | 0.51                   | $6.93 \pm 2.90$                                                     |
|       | 0 – 349 (unit A1)     | 6                                       | 1214                                 | 0.44                   | $5.00\pm2.55$                                                       |
|       | 349 – 925 (unit A2)   | 6                                       | 998                                  | 0.05*                  | $0.50 \pm 0$                                                        |
|       | 925 – 1210 (unit A3)  | 4                                       | 1299                                 | 0.05*                  | $0.66\pm0.01$                                                       |
|       | 1210 – 1980 (unit A4) | 12                                      | 1377                                 | 0.83                   | $11.03 \pm 1.62$                                                    |
|       | total core            | 28                                      | 1250                                 | 0.47                   | $6.07 \pm 1.80$                                                     |